# Sampling Binary Data by Denoising through Score Functions

**Francis Bach** [1]    **Saeed Saremi** [2]

## Abstract

Gaussian smoothing combined with a probabilistic framework for denoising via the empirical Bayes formalism, i.e., the Tweedie-Miyasawa formula (TMF), are the two key ingredients in the success of score-based generative models in Euclidean spaces. Smoothing holds the key for easing the problem of learning and sampling in high dimensions, denoising is needed for recovering the original signal, and TMF ties these together via the score function of noisy data. In this work, we extend this paradigm to the problem of learning and sampling the distribution of binary data on the Boolean hypercube by adopting Bernoulli noise, instead of Gaussian noise, as a smoothing device. We first derive a TMF-like expression for the optimal denoiser for the Hamming loss, where a score function naturally appears. Sampling noisy binary data is then achieved using a Langevin-like sampler which we theoretically analyze for different noise levels. At high Bernoulli noise levels sampling becomes easy, akin to log-concave sampling in Euclidean spaces. In addition, we extend the sequential multi-measurement sampling of Saremi et al. (2024) to the binary setting where we can bring the "effective noise" down by sampling multiple noisy measurements at a fixed noise level, without the need for continuous-time stochastic processes. We validate our formalism and theoretical findings by experiments on synthetic data and binarized images.

## 1. Introduction

We would like to draw samples from a distribution $p$ on the Boolean hypercube $\{-1, 1\}^d$. Langevin Markov

chain Monte Carlo (MCMC) is a general-purpose class of gradient-based algorithms for sampling from a distribution $\mu$ in the Euclidean space $\mathbb{R}^d$, whose convergence properties are studied extensively with the assumption that $\mu$ is log-concave (Dalalyan, 2017; Durmus & Moulines, 2017; Cheng et al., 2018; Chewi, 2024). Recently, Gaussian smoothing was effectively used for mapping the general problem of sampling in Euclidean space to log-concave sampling (Saremi et al., 2024). Inspired by this line of work, we approach the problem of sampling binary data with a "smoothing philosophy," where Bernoulli noise plays a prominent role.

In Euclidean space one can ease the sampling problem by opting for sampling noisy data. In particular, instead of the random variable $x$, we opt for sampling the random variable $y = x + \varepsilon$, where $\varepsilon \sim \mathcal{N}(0, \sigma^2 I)$ follows a Gaussian distribution. This scheme involves a single hyperparameter, the standard deviation $\sigma$. Algebraically, this is akin to sampling from the smoother density $\nu_\sigma$ of $y$, which is the convolution of the distribution $\mu$ of $x$ with the Gaussian distribution. From a geometric perspective, the noise effectively "fills up" the space with probability mass (the degree of which one controls with $\sigma$) thus navigating the space becomes easier. One can then "clean up" the mass that is added to the space using denoising. In particular, classical results in statistics (Robbins, 1956; Miyasawa, 1961) state:

$$\mathbb{E}[x|y] = y + \sigma^2 \nabla \log \nu_\sigma(y),$$

which we refer to as the Tweedie-Miyasawa formula (TMF). Note that $\mathbb{E}[x|y]$ is the least-squares estimator of clean data $x$ given a noisy observation $y$, and $\nabla \log \nu_\sigma$ is known as the score function (Hyvärinen, 2005).

In the generative modeling setting, where the distribution is unknown but we have access to data $\{x^{(i)}\}_{i=1}^n$, one can turn TMF into a supervised least-squares denoising objective for learning the score function of noisy data, where the noisy data $y = x + \varepsilon$, $\varepsilon \sim \mathcal{N}(0, \sigma^2 I)$ is the input and the clean data $x$ is the target (Hyvärinen, 2005; Vincent, 2011; Saremi & Hyvärinen, 2019). One can then use Langevin MCMC ("walk") to sample from the learned $\nu_\sigma(y)$; the noisy samples can be cleaned up with the learned denoiser ("jump"). This sampling scheme was referred to as the walk-jump sampling which we denote by WJS-1 ("1" anticipates the extension we discuss below). There is a clear trade-off

[1]Inria, Ecole Normale Supérieure, PSL Research, University, Paris, France [2]Frontier Research, Prescient Design, Genentech. Correspondence to: Francis Bach <francis.bach@inria.fr>, Saeed Saremi <saremi.saeed@gene.com>.

*Proceedings of the $42^{nd}$ International Conference on Machine Learning*, Vancouver, Canada. PMLR 267, 2025. Copyright 2025 by the author(s).

here: for higher $\sigma$, sampling from $\nu_\sigma(y)$ becomes easier, but the distribution of denoised samples itself goes farther away from $\mu(x)$. Despite this trade-off, WJS-1 has proven to be effective in some applications (Pinheiro et al., 2023; Frey et al., 2024; Kirchmeyer et al., 2024).

The sampling trade-off in WJS-1 is addressed in multi-measurement models (Saremi & Srivastava, 2022; Saremi et al., 2024), in which one is interested in the distribution $\nu_\sigma(y_{1:m})$ associated with $y_{1:m} := (y_1, \ldots, y_m)$, where $y_k = x + \varepsilon_k$, $k \in [m]$, and $\varepsilon_k \sim \mathcal{N}(0, \sigma^2 I)$ all independent and independent of $x$. Saremi et al. (2024) studied a sequential strategy for sampling from $\nu_\sigma(y_{1:m})$ and showed that the noise level effectively goes down (as far as the denoiser is concerned) at the rate $\sigma/\sqrt{m}$. Furthermore, if one chooses $\sigma$ such that the distribution $\nu_\sigma(y_1)$ is log-concave, all the subsequent conditional distributions $\nu_\sigma(y_k|y_{1:k-1})$, $k \in [m]$ remain log-concave. The general sampling problem is therefore mapped to a sequence of log-concave sampling while the effective noise goes down via this accumulation of measurements. We refer to this scheme as WJS-$m$, which involves two hyperparameters: the noise level $\sigma$, and the number of measurements $m$. This approach has deep connections to sampling via diffusion and stochastic localization (Montanari, 2023). The main conceptual difference is that the multi-measurement sampling does not involve discretizing an SDE (Song et al., 2021; Campbell et al., 2022) for bringing the noise down. Fundamentally, this is due to the discrete nature of measurement accumulation.

### 1.1. Contributions

Given this background, we approached the problem of sampling from a distribution $p(x)$ on $\{-1, 1\}^d$ by devising a smoothing method, with the key restriction to stay on the Boolean hypercube (in this purely binary world Gaussian noise does not exist). The natural choice to "smooth" the binary data is to use (isotropic) random sign flips dictated by the Bernoulli noise: $y = x \circ \varepsilon$, where $\circ$ denotes the Hadamard (i.e., pointwise) product. The noise $\varepsilon$ is drawn from the Bernoulli distribution, $\mathbb{P}(\varepsilon_i = 1) = \sigma(2\alpha)$, where $\sigma$ is the sigmoid function, and $\alpha \geqslant 0$ is the noise parameter. The probability mass function of the noisy data $q_\alpha(y)$ happens to be a transformation of $p(x)$ via an exponential tilt governed by $\exp(\alpha x^\top y)$. As $\alpha$ decreases, the probability mass gets more spread out on the hypercube, thus easing the sampling problem, where in the extreme case, $\alpha = 0$, we arrive at the uniform distribution.

For denoising there are subtle differences between the Boolean/Bernoulli and Euclidean/Gaussian setups, where the optimal denoiser $f$ takes values on the Boolean hypercube and the Hamming loss is the natural loss. We show in Lemma 2.1 that $f$ takes the form $f(y) = \text{sign}(\mathbb{E}[x|y])$, and

in Lemma 2.2 we show that

$$\mathbb{E}[x|y] = \frac{1}{\alpha} \nabla \log q_\alpha(y).$$

This is essentially the form of TMF for the Bernoulli noise, where crucially the score function appears again. We should emphasize that the score function is well-defined here since $q_\alpha(y)$ has an analytical form (dictated by the exponential tilt) beyond $\{-1, 1\}^d$. Finally, similar to the Gaussian case discussed earlier we can learn the score function given a dataset $\{x^{(i)}\}_{i=1}^n$ by denoising, the subtle difference here is that since $x$ and $y$ are both binary, we can also set up the denoising objective via logistic regression (Section 2.3). Naturally, denoising becomes harder as $\alpha$ decreases, which we can characterize by the Wasserstein distance between the law of $x$ and the law of $\mathbb{E}[x|y]$ (Lemma 2.3).

Our second main contribution in this work is to analytically study sampling from $q_\alpha(y)$ using gradient-based methods with a formal understanding of the role the Bernoulli noise level $\alpha$ plays in easing the problem of sampling binary data. There has been a recent interest on devising gradient-based sampling strategies for discrete distributions from the perspective of Gibbs sampling (Grathwohl et al., 2021), and Langevin MCMC (Zhang et al., 2022). Our approach here is close to the later, where in addition we introduce a new *two-stage* discrete Langevin MCMC algorithm (Section 3.2), with improved behavior at high noise.

Langevin-like updates are especially motivated here on two fronts: (i) the probability mass of noisy data is more spread out on the Boolean hypercube and it demands coming up with Markov moves where many coordinates are updated in parallel (in contrast to "cautious" single-coordinate Gibbs updates), (ii) the score function $\nabla \log q_\alpha(y)$ is readily accessible to be used via denoising and our binary TMF. Regarding the first point, we theoretically analyze the contraction properties of the vanilla (one-stage) and two-stage discrete Langevin MCMC, where the noise level $\alpha$ plays a prominent role in the exponential convergence of the algorithms. To our knowledge, there is no prior work on the exponential convergence of discrete Langevin-like algorithms in the Wasserstein metric (Propositions 3.1 and 3.3). Furthermore, we extend our contraction results by proving bounds on the distance between the stationary distributions of the discrete Langevin algorithms and the target distribution $q_\alpha$ in the Wasserstein metric (Propositions 3.2 and 3.4). These results again highlight the important role the noise level $\alpha$ plays.

Informally, there are parallels between contractivity results for high Bernoulli noise (small $\alpha$) and the exponential convergence of Langevin MCMC for log-concave distributions achieved for large $\sigma$ in the Euclidean/Gaussian case (Saremi et al., 2024, Theorem 1). We make this connection formal from the angle of sampling multiple noisy data, where multiple Bernoulli noise is added independently to clean

data, where the noise level is held fixed. The TMF for the multiple Bernoulli measurements takes the form

$$\mathbb{E}[x|y_{1:m}] = \frac{1}{m\alpha}\nabla \log q_{m\alpha}(\bar{y}_{1:m}),$$

where $\bar{y}_{1:m} = \frac{1}{m}\sum_{i=1}^{m} y_i$, which corresponds to a reduced noise dictated by $m\alpha$ (Lemma 2.4).

We conduct a set of experiments on synthetic data, where we study a mixture model on $\{-1,1\}^d$, akin to mixture of Gaussians in $\mathbb{R}^d$. The experiments were designed to quantify denoising for strong and weak priors and probe the sampling properties of our scheme. We also conduct experiments on binarized MNIST by qualitatively studying the role of $\alpha$, and demonstrate the fast mixing our algorithm can achieve with essentially no tuning (the step-size is simply set to $1/\alpha$).

### 1.2. Related work

There is a growing body of work on sampling from discrete distributions with score-based models that build on denoising diffusion models (Sohl-Dickstein et al., 2015; Hoogeboom et al., 2021; Austin et al., 2021; Campbell et al., 2022; Lou et al., 2024; Shi et al., 2024; Kim et al., 2025). There are variations between these models, but they are all fundamentally formulated based on a forward/backward continuous-time diffusion process for corrupting the data and learning score functions, via denoising, to reverse the process. In particular, Montanari (2023, Section 4.4) considers a noise process similar to ours and Pham et al. (2025) show how it can be inverted using Markov jump processes, with denoisers which are similar to the ones defined in Section 2. Algorithmically, these continuous-time processes are then discretized using various schemes. Our approach here is fundamentally different with a single noise scale sampling strategy: at each stage of measurement accumulation the data is sampled at a fixed noise scale. The process to bring the noise down is therefore discrete by nature, characterized by a single hyperparameter, the number of measurements $m$, in contrast to devising a noise schedule in diffusion-based prior works.

## 2. Denoising and Binary Score Functions

We consider a binary random vector $x \in \{-1,1\}^d$, with probability mass function $p(x)$ (that sums to one).

### 2.1. Noise models for binary vectors

A natural noise model is to use random sign flips, that is,

$$y = x \circ \varepsilon \qquad (1)$$

(for the component-wise product $\circ$), where $\varepsilon \in \{-1,1\}^d$ has independent components, and, for $i \in \{-1,\dots,1\}^d$,

$$\mathbb{P}(\varepsilon_i = 1) = \sigma(2\alpha),$$

where $\sigma(u) = \frac{1}{1+e^{-u}} = \frac{e^{u/2}}{e^{u/2}+e^{-u/2}}$ is the sigmoid function, and $\alpha \geqslant 0$ is the noise parameter (the noise *decreases* with $\alpha$). When $\alpha$ is large, $\sigma(2\alpha)$ is close to one, and thus $\varepsilon$ is the vector of all ones with high probability, and $y$ is close to $x$ (small noise). When $\alpha$ is equal to zero, then $\sigma(2\alpha) = 1/2$, and $\varepsilon$ is uniform, and so is $y$ (high noise). Moreover, The expected number of flips is equal to $d\sigma(-2\alpha)$, and goes from $d/2$ when $\alpha = 0$ to 0 exponentially fast when $\alpha = +\infty$. We refer to this noise model as Bernoulli noise (in $\{-1,1\}^d$).

We can write the probability mass $r$ function of $\varepsilon$ as

$$r(\varepsilon) = \prod_{i=1}^{d} \sigma(2\alpha\varepsilon_i) = \prod_{i=1}^{d} \frac{e^{\alpha\varepsilon_i}}{e^{\alpha}+e^{-\alpha}} = \frac{1}{(2\cosh\alpha)^d} e^{\alpha 1_d^\top \varepsilon},$$

and the probability mass function $q_\alpha$ of $y$ defined in Eq. (1) as:

$$
\begin{aligned}
q_\alpha(y) &= \sum_{x \in \{-1,1\}^d} p(x) r(y \circ x) \\
&= \frac{1}{(2\cosh\alpha)^d} \sum_{x \in \{-1,1\}^d} p(x) e^{\alpha x^\top y} \qquad (2) \\
&= \sigma(2\alpha)^d \sum_{x \in \{-1,1\}^d} p(x) e^{-\frac{\alpha}{2}\|x-y\|_2^2},
\end{aligned}
$$

since on the hypercube $\|x\|_2^2 = \|y\|_2^2 = d$.

When $\alpha = 0$, $q_\alpha$ is the uniform distribution, while for $\alpha = +\infty$, $q_\alpha = p$. Thus, $\alpha$ plays exactly the role of the *inverse* variance, as can be seen with last expression above that mimics Gaussian noise.

A key observation is that the function $q_\alpha(y)$ defined in Eq. (2) is defined for *all* $y \in \mathbb{R}^d$, and not only in $\{-1,1\}^d$, so that we can take continuous gradients—not discrete gradients as sometimes done for score matching extensions (Hyvärinen, 2007; Meng et al., 2022).

### 2.2. Denoising

Given the noisy (random) version $y \in \{-1,1\}^d$, how can we recover a good denoised $x \in \{-1,1\}^d$? Like for the Gaussian case, once given a loss function, the optimal denoiser has a closed-form expression. We consider the Hamming loss, which has several expressions when $x, x' \in \{-1,1\}^d$, as an $\ell_1$-norm or a squared $\ell_2$-norm:

$$
\begin{aligned}
\ell(x,x') &= \sum_{i=1}^{d} 1_{x_i \neq x_i'} = \frac{1}{2}\sum_{i=1}^{d}|x_i - x_i'| = \frac{1}{2}\|x - x'\|_1 \\
&= \frac{1}{4}\sum_{i=1}^{d}|x_i - x_i'|^2 = \frac{1}{4}\|x - x'\|_2^2.
\end{aligned}
$$

It simply counts the number of mistakes, between 0 and $d$. We then obtain the optimal denoiser from the conditional expectation (which extends classical results from binary classification, see Bach (2024, Section 4.1)).

**Lemma 2.1** (Optimal denoiser). *Given a joint distribution on $(x,y)$, the function $f : \{-1,1\}^d \rightarrow \{-1,1\}^d$ that minimizes $\mathbb{E}[\ell(x, f(y))]$ is $f(y) = \text{sign}(\mathbb{E}[x|y])$.*

*Proof.* We have:

$$\mathbb{E}[\ell(x, f(y))] = \sum_{y \in \{-1,1\}^d} p(y) \sum_{x \in \{-1,1\}^d} p(x|y)\ell(x, f(y)),$$

and $f_i(y) \in \{-1, 1\}$ can be optimized independently for each $y \in \{-1, 1\}^d$ and $i \in \{1, \dots, d\}$, and maximizes $\sum_{x \in \{-1,1\}^d} p(x|y)1_{x_i=f_i(y)} = \mathbb{P}(x_i = f_i(y)|y)$. Thus, $f_i(y) = 1$ if $\mathbb{P}(x_i = 1|y) > \mathbb{P}(x_i = -1|y)$, which exactly leads to the sign of $\mathbb{E}[x_i|y]$. The value of the sign at zero can be taken to be uniformly at random in $\{-1, 1\}$. $\square$

We can now consider the noise model in Eq. (2) from Section 2.1 and compute the gradient of $\log q_\alpha$ as

$$\nabla \log q_\alpha(y) = \frac{\sum_{x \in \{-1,1\}^d} p(x)\alpha x e^{\alpha x^\top y}}{\sum_{x \in \{-1,1\}^d} p(x)e^{\alpha x^\top y}}, \quad (3)$$

which is exactly $\alpha\mathbb{E}[x|y]$, leading to the following lemma.

**Lemma 2.2** (Denoising through score functions). *For the function $q_\alpha$ defined in Eq. (2) for all $y \in \mathbb{R}^d$ that characterizes the random sign flip model, we have:*

$$\mathbb{E}[x|y] = \frac{1}{\alpha}\nabla \log q_\alpha(y).$$

We refer to the function $\nabla \log q_\alpha(y)$ as the score function. It allows to obtain the optimal denoiser by taking the sign. This denoiser has a performance that degrades when $\alpha$ goes to zero. We consider the Wasserstein distance derived from the loss $\ell$, that is, given two distributions $p$ and $q$ on $\{-1, 1\}^d$, we consider $W(p, q)$ as the minimum expectation $\mathbb{E}[\ell(x, y)]$ over all distributions on $(x, y)$ with marginals $p$ on $x$ and $q$ on $y$ (Peyré & Cuturi, 2019). The following lemma provides an upper-bound that extends the Gaussian result from Saremi et al. (2024).

**Lemma 2.3** (Denoising performance). *For the noise model defined in Eq. (2), we have:*

$$W(\text{law of } x, \text{law of } \text{sign}(\mathbb{E}[x|y])) \leqslant de^{-2\alpha}.$$

*Proof.* We consider the natural coupling with $y = x \circ \varepsilon$ where $\varepsilon$ is independent of $x$ and $\varepsilon$ has independent components, and simply use the fact that $\mathbb{E}[\ell(x, f(y))]$ is minimized exactly by $f(y) = \text{sign}(\mathbb{E}[x|y])$, and thus is less than the loss of the naive denoiser that simply outputs $y$, for which $\mathbb{E}[\ell(x, y)] = d\sigma(-2\alpha)$, which is less than $de^{-2\alpha}$. $\square$

Like for the Gaussian case, this bound is true regardless of the strength of the prior on $x$. If $p(x)$ is uniform, it cannot really be improved. However, when the prior is strong, better bounds could be derived.

Note that as opposed to Gaussian noise, the denoising performance goes exponentially to zero when $\alpha$ grows.

### 2.3. Learning the score function

In order to learn the denoiser, it is natural to consider observations $x^{(1)}, \dots, x^{(n)} \in \{-1, 1\}^d$, generate independent noise variables $\varepsilon^{(1)}, \dots, \varepsilon^{(n)} \in \{-1, 1\}^d$ (with $\mathbb{P}(\varepsilon_j^{(i)} = 1) = \sigma(2\alpha)$), and parameterize a denoiser $\mathbb{E}[x|y] = 2\sigma(f_\theta(y)) - 1 = \tanh \frac{f_\theta(y)}{2} \in \mathbb{R}^d$, with thus $f_\theta(y)$ of the form $2\text{atanh}\left[\frac{1}{\alpha}\nabla \log q_\alpha(y)\right]$.

We can learn it through the following denoising criterion (which is exactly logistic regression):

$$\frac{1}{n}\sum_{i=1}^{n}\sum_{j=1}^{d} \log\left(1 + \exp\left(-x_j^{(i)}f_\theta(x^{(i)} \circ \varepsilon^{(i)})_j\right)\right).$$

We could also use a least-squares objective, $\frac{1}{n}\sum_{i=1}^{n}\sum_{j=1}^{d}\left|x_j^{(i)} - g_\theta(x^{(i)} \circ \varepsilon^{(i)})_j\right|^2$, where the optimal $g(y)$ is $\mathbb{E}[x|y]$.

### 2.4. Multiple measurements

Following Saremi et al. (2024), if we assume that we have $m$ measurements $y_1, \dots, y_m \in \{-1, 1\}^d$ obtained by adding independent noises to the same $x$, then we have from Eq. (3):

$$\mathbb{E}[x|y_1, \dots, y_m] = \frac{1}{m\alpha}\nabla \log q_{m\alpha}(\bar{y}_{1:m}), \quad (4)$$

with $\bar{y}_{1:m} = \frac{1}{m}(y_1 + \cdots + y_m) \in [-1, 1]^d$. Note that this requires to know the function $\nabla \log q_{m\alpha}$ beyond $\{-1, 1\}^d$.

Since the denoiser has to "work" for $y \notin \{-1, 1\}^d$, and for noise levels $m\alpha$, to learn the score function, we can simply generate multiple measurements and average the corresponding $y$'s (in the Gaussian case, this was possible directly by adding a noise with variance divided by $m$), and use the same denoising objective as Section 2.3 (we can also directly sample multinomial random variables to avoid generating $m$ measurements).

**Denoising performance.** We can extend the denoising performance result when given $m$ measurements by studying the sign of $y_1 + \cdots + y_m$. This is uniquely defined as soon as $m$ is odd, and when $m$ is even, and $y_1 + \cdots + y_m$ is equal to zero, we output $-1$ or $1$ with equal probabilities. We can then extend Lemma 2.3 (see the proof based on Chernoff's bound in Appendix A).

**Lemma 2.4** (Denoising performance, multiple measurements). *For the noise model defined in Eq. (2) and $m$ measurements, we have:*

$$W(\text{law of } x, \text{law of } \text{sign}(\mathbb{E}[x|y_1, \dots, y_m])) \leqslant de^{-mH(\alpha)},$$

*with $H(\alpha) = \frac{1}{2}\log\frac{1+\cosh 2\alpha}{2} > 0$ for $\alpha > 0$, is equivalent to $\alpha^2/2$ for $\alpha$ tends to zero, and to $\alpha - \log 2$ when $\alpha$ tends to infinity.*

**Score functions for sequential sampling.** Extending the Gaussian case, multiple measurement can be efficiently sampled *sequentially*. If we want to sample $y_m$ given $y_1, \ldots, y_{m-1}$ using score functions, we have the joint probability mass function:

$$p(y_1, \ldots, y_m) = \sum_{x \in \{-1,1\}^d} p(x) \frac{e^{\alpha x^\top (y_1 + \cdots + y_m)}}{(2 \cosh \alpha)^{dm}},$$

with
$$\begin{aligned}
&\nabla_{y_m} \log p(y_1, \ldots, y_m) \\
&= \alpha \frac{\sum_{x \in \{-1,1\}^d} p(x) x e^{\alpha x^\top (y_1 + \cdots + y_m)}}{\sum_{x \in \{-1,1\}^d} p(x) e^{\alpha x^\top (y_1 + \cdots + y_m)}} \\
&= \frac{1}{m} \nabla \log q_{m\alpha}(\bar{y}_{1:m}),
\end{aligned}$$

which depends on the score function with parameter $m\alpha$, taken at $\bar{y}_{1:m} \in [-1, 1]^d$ (as mentioned earlier we are able to learn a score functions for elements in the interior of the hypercube).

## 3. Sampling with Discrete Langevin

We need to sample from the model in Eq. (2), for a probability mass function $q$ which is defined not only on all $y \in \{-1, 1\}^d$ but on $\mathbb{R}^d$, for which we only know $s(y) = \nabla \log q(y)$. This is the same for conditional sampling from Section 2.4.

**Assumptions.** Note that the density $q$ is uniquely defined up to a constant for vertices of the hypercube, but that there are multiple versions on the whole hypercube. In particular, one can add to the score $s$ any linear function (and there are additional invariances).

We will use the following regularity conditions on $s$ that are satisfied by $q$ defined in Eq. (2), that is, for all $y, y' \in \mathbb{R}^d$,

$$\|s(y)\|_\infty \leqslant \beta_1, \text{ and } \|s(y) - s(y')\|_\infty \leqslant \beta_2 \|y - y'\|_1. \quad (5)$$

These assumptions are akin to regularity assumptions made in optimization on the gradient functions, here adapted to the binary case. For the function $q$ defined in Eq. (2) and the associated $s = \nabla \log q$, we have $\beta_1 = \alpha$ and $\beta_2 = \alpha^2$ (this is a direct consequence of taking another derivative in Eq. (3), leading to $\nabla^2 \log q_\alpha(y) = \alpha^2 \mathrm{cov}(x|y)$). The same bounds hold for sequential sampling from Section 2.4 (thus benefiting from the same speed as a small $\alpha$ while denoising has the same performance as $m\alpha$).

### 3.1. One-stage discrete Langevin sampler

In order to obtain an approximate sample from $q$, we consider the following Markov transition kernel proposed

---

**Algorithm 1** The single-measurement one-stage discrete Langevin algorithm.

1: **Parameter:** noise level $\alpha$
2: **Input:** steps count $n$, step size $\eta$, score function $s_\alpha$
3: **Output:** $\hat{X}$
4: Initialize $Y_0 \in \{-1, 1\}^d$
5: **for** $i = 1$ to $n$ **do**
6:      $Z \leftarrow Y_{i-1}$
7:      $\varepsilon \leftarrow \mathrm{Bernoulli}(\sigma(2/\eta + Z \circ s_\alpha(Z))) \in \{-1, 1\}^d$
8:      $Y_i \leftarrow Z \circ \varepsilon$
9: **end for**
10: **return** $\hat{X} = \mathrm{sign}\left(s_\alpha(Y_n)/\alpha\right)$

---

by Zhang et al. (2022), which is adapted to log-probability-mass functions that are defined on $\mathbb{R}^d$:

$$\begin{aligned}
t(y'|y) &\propto \exp\left(\tfrac{1}{2} s(y)^\top (y' - y) - \tfrac{1}{2\eta} \|y' - y\|_2^2\right) \\
&\propto \exp\left(\left(\tfrac{1}{2} s(y) + \tfrac{1}{\eta} y\right)^\top y'\right), \quad (6)
\end{aligned}$$

which given $y$, has independent components for $y'$. Note that without the constraint that $y' \in \{-1, 1\}^d$, the first expression above becomes $y' = y + \frac{\eta}{2} s(y) + \mathcal{N}(0, \eta I)$, i.e., *exactly* (Gaussian) Langevin MCMC.

As opposed to Gaussian Langevin, even with a vanishing step-size $\eta$, the stationary distribution of this Markov chain may not approach $q$. We can however prove convergence results that are adapted to our situation.

**Convergence results.** We now provide two propositions characterizing the convergence of the Markov chain defined in Eq. (6); see Algorithm 1. The first proposition below implies an exponential convergence of the Markov chain (which is not studied by Zhang et al. (2022)). See the proof in Appendix B, based on standard contraction arguments.

**Proposition 3.1** (Contractivity). *Assume regularity conditions in Eq. (5) with $4\beta_2 d e^{2\beta_1} \leqslant 1$. Given $y, z \in \{-1, 1\}^d$, we have, for the transition kernel defined in Eq. (6):*

$$W(t(\cdot|y), t(\cdot|z)) \leqslant \left(1 - \tfrac{1}{2} e^{-\frac{2}{\eta} - \beta_1}\right) \ell(y, z).$$

We note that when $\beta_1 = \alpha$ and $\beta_2 = \alpha^2$, the constraint becomes $4\alpha^2 d e^{2\alpha} \leqslant 1$ and is satisfied as soon as $\alpha \leqslant \frac{1}{4\sqrt{d}}$ (since then we have $e^{2\alpha} \leqslant 2$), without any assumption about the distribution we want to sample from.

From the previous proposition, we deduce that for each $\eta > 0$, the Markov chain always converge exponentially fast to the unique stationary distribution for the Wasserstein distance (Levin & Peres, 2017). Given the exponential rate in $1 - \frac{1}{2} e^{-2/\eta - \alpha}$, we can choose a step-size $\eta = 1/\alpha$ without losing too much in mixing time (this extends the strategy of Saremi & Hyvärinen (2019) in the Gaussian case,

**Algorithm 2** The single-measurement two-stage discrete Langevin algorithm.

1: **Parameter:** noise level $\alpha$
2: **Input:** steps count $n$, step size $\eta$, score function $s_\alpha$
3: **Output:** $\hat{X}$
4: Initialize $Y_0 \in \{-1, 1\}^d$
5: **for** $i = 1$ to $n$ **do**
6:    $\varepsilon \leftarrow \mathrm{Bernoulli}(\sigma(2/\eta)) \in \{-1, 1\}^d$
7:    $Z \leftarrow Y_{i-1} \circ \varepsilon$
8:    $\varepsilon \leftarrow \mathrm{Bernoulli}(\sigma(2/\eta + 2Z \circ s_\alpha(Z))) \in \{-1, 1\}^d$
9:    $Y_i \leftarrow Z \circ \varepsilon$
10: **end for**
11: **return** $\hat{X} = \mathrm{sign}\left(s_\alpha(Y_n)/\alpha\right)$

where the step-size for the Langevin algorithm is taken to be the noise variance).

We now show that the stationary distribution of the Markov chain defined by the transition kernel $t$ cannot be too far from the one of $q$. See the proof based on comparing to Metropolis-Hasting steps in Appendix C. Note that Zhang et al. (2022) only study the situations where the log-density is quadratic (or close to quadratic) without explicit constants, and without results on mixing time.

**Proposition 3.2** (Distance to stationary distribution). *Assume regularity conditions in Eq. (5) with $4\beta_2 d e^{2\beta_1} \leqslant 1$. Let $q'$ be the stationary distribution of the transition kernel defined in Eq. (6). Then,*

$$W(q', q) \leqslant 2d\left(2d\beta_1 e^{2\beta_1} + \sqrt{d\beta_1 e^{2\beta_1}}\right). \qquad (7)$$

In our situation where $\beta_1 = \alpha$ and $\beta_2 = \alpha^2$, this is small compared to the diameter $d$ of the hypercube (its maximal value) only when $\alpha$ is small compared to $1/d$. Note that the step-size $\eta$ does not appear directly in the bound in Eq. (7), but if it is too small, because of Prop. 3.1 the mixing time will be large (this also applies to Prop. 3.4).

### 3.2. Two-stage Langevin sampler

The default Langevin sampler does not have the nice property that if $q(y) \propto e^{s^\top y}$ for some $s \in \mathbb{R}^d$ (i.e., independent components), then the stationary distribution is exact.

Following the continuous-space algorithms from Lee et al. (2021); Chewi (2024), we consider instead the idealized two-stage sampler defined below, which is Gibbs sampling for the joint model on $(y, z)$ in $\{-1, 1\}^d \times \{-1, 1\}^d$:

$$q(y, z) = q(y)\frac{1}{(2\cosh\frac{1}{\eta})^d}e^{\frac{1}{\eta}y^\top z} \propto q(y)e^{\frac{1}{\eta}y^\top z},$$

for which the conditional distributions can be computed as

$$
\begin{aligned}
q(z|y) &= \frac{1}{(2\cosh\frac{1}{\eta})^d}e^{\frac{1}{\eta}y^\top z} \\
q(y|z) &\propto q(y)e^{\frac{-1}{2\eta}\|y-z\|_2^2},
\end{aligned}
$$

which we approximate by expanding $\log q(y) \approx \log q(z) + s(z)^\top(y - z)$, leading to:

$$
\begin{aligned}
u(z|y) &= q(z|y) = \frac{1}{(2\cosh\frac{1}{\eta})^d}e^{\frac{1}{\eta}y^\top z} & (8) \\
u(y|z) &\propto q(z)e^{\nabla \log q(z)^\top(y-z)}e^{\frac{-1}{2\eta}\|y-z\|_2^2} \\
&\propto e^{y^\top(\frac{z}{\eta}+s(z))}. & (9)
\end{aligned}
$$

Thus the Markov chain (with transition kernel $v$) defined by Gibbs sampling to go from $y$ to $y'$ is defined as follows: take $z$ with a random flip with probability $1 - \sigma(2/\eta)$, and then perform independent (non-uniform) flips with probability $1 - \sigma(2/\eta + 2z_i s(z)_i)$ to obtain $z'$; see Algorithm 2.

Note that without the constraint that $y' \in \{-1, 1\}^d$, the overal update becomes $y' = y + \eta s(y) + \mathcal{N}(0, 2\eta I)$, i.e., *exactly* (Gaussian) Langevin MCMC with a step-size twice bigger than the single-stage sampler.

**Convergence results.** When $\log q(y)$ is linear in $y$, then the proposals defined by $u$ are equal to the ones defined by $q$ (and thus the stationary distribution is exact). Otherwise, we can show the following contractivity result (see the proof based on contraction arguments in Appendix D).

**Proposition 3.3** (Contractivity, two-stage sampler). *Assume regularity conditions in Eq. (5) with $8d\beta_2 e^{4\beta_1} \leqslant 1$. Given $y^{(1)}, y^{(2)} \in \{-1, 1\}^d$, we have, for the transition kernel $v$ defined in Eq. (8) and Eq. (9):*

$$W(v(\cdot|y^{(2)}), v(\cdot|y^{(1)})) \leqslant \left(1 - \frac{1}{2}e^{-\frac{2}{\eta}-2\beta_1}\right)\ell(y^{(1)}, y^{(2)}).$$

Like in Section 3.1, this leads to exponential convergence to a unique stationary distribution. We can now look at the distance between the stationary distribution of the Markov chain and $q$. We make the assumption that the score $s$ that we use satisfies the usual inequality of convex smooth functions (Bach, 2024, Section 5.2), that is, for all $y, z \in \{-1, 1\}^d$,

$$0 \leqslant \log q(y) - \log q(z) - s(z)^\top(y-z) \leqslant \frac{\beta_2}{2}\|y-z\|_1^2, \quad (10)$$

which is satisfied by $s(y) = \nabla \log q(y)$ in Eq. (2). See the proof of Prop. 3.4 based on comparing to Metropolis-Hasting steps in Appendix E.

**Proposition 3.4** (Distance to stationary distribution). *Assume regularity conditions in Eq. (5) and Eq. (10) with $8\beta_2 d e^{4\beta_1} \leqslant 1$. Assume $e^{-2/\eta+2\beta_1} \leqslant \frac{1}{d}$. Let $q'$ be the stationary distribution of the two-stage sampler. Then,*

$$W(q', q) \leqslant 12d\sqrt{\beta_2 d}.$$

**Algorithm 3** Multi-measurement binary sampling via single-stage discrete Langevin algorithm (Alg. 1) in the inner loop.

1: **Parameter:** noise level $\alpha$
2: **Input:** number of measurements $m$, number of steps per measurement $n$, score functions $\{s_\alpha, \ldots, s_{m\alpha}\}$.
3: **Output:** $\hat{X}$
4: Initialize $\overline{Y} \leftarrow 0$
5: **for** $t = 1$ to $m$ **do**
6:     Initialize $Y_0 \in \{-1, 1\}^d$
7:     **for** $i = 1$ to $n$ **do**
8:         $Z \leftarrow \overline{Y} + (Y_{i-1} - \overline{Y})/t$
9:         $\varepsilon \leftarrow \text{Bernoulli}(\sigma(2/\eta + Y_{i-1} \circ s_{t\alpha}(Z)/t))$
10:        $Y_i \leftarrow Y_{i-1} \circ \varepsilon$
11:     **end for**
12:     $\overline{Y} \leftarrow \overline{Y} + (Y_n - \overline{Y})/t$
13: **end for**
14: **return** $\hat{X} = \text{sign}\left(s_{m\alpha}(\overline{Y})/(m\alpha)\right)$

Moreover, as shown in Appendix E, if the step-size $\eta$ is small enough, we get a dependence in $d \cdot d\beta_2$.

For our problem where $\beta_1 = \alpha$ and $\beta_2 = \alpha^2$, the constraint in $\eta$ leads to a mixing time proportional to $d$, but to a distance to the true distribution $q$ proportional to $d \cdot \alpha^2 d$ or $d \cdot \alpha \sqrt{d}$ as opposed to $d \cdot \sqrt{\alpha} d$ for the one-stage sampler, thus an advantage for small $\alpha$ (high noise), where we only need $\alpha \ll 1/\sqrt{d}$ instead of $\alpha \ll 1/d$ for one-stage sampling.

For completeness, we also provide the multi-measurement binary sampling algorithm in Alg. 3.

## 4. Comparison with Gaussian Noise

An alternative is to add Gaussian noise and define

$$y_\text{G} = x + \varepsilon, \ \varepsilon \sim \mathcal{N}(0, \tfrac{1}{\alpha}I).$$

We then have $\mathbb{E}[x|y_G] = y_G + \frac{1}{\alpha}\nabla \log q_\alpha^\text{G}(y_G)$, from the classical Tweedie-Miyasawa formula, with $q_\alpha^\text{G}$ the density of $y_G$:

$$q_\alpha^\text{G}(y_\text{G}) \propto \sum_{x \in \{-1,1\}^d} p(x) e^{\alpha x^\top y} e^{-\frac{\alpha}{2}\|y\|_2^2} e^{-\frac{\alpha}{2}\|x\|_2^2}$$

$$\propto \sum_{x \in \{-1,1\}^d} p(x) e^{\alpha x^\top y} e^{-\frac{\alpha}{2}\|y\|_2^2} \propto q_\alpha(y_\text{G}) e^{-\frac{\alpha}{2}\|y\|_2^2},$$

where $q_\alpha$ is the density of the binary case defined in Eq. (2). Thus,

$$\mathbb{E}[x|y_\text{G}] = \tfrac{1}{\alpha}\nabla \log q_\alpha(y_\text{G}),$$

that is, the exact same denoiser function (now applied to an element of $\mathbb{R}^d$ and not $\{-1, 1\}^d$).

In terms of denoising performance, for the same $\alpha$, we see in our experiments that they behave similarly. However, in terms of mixing time of Langevin, for the Gaussian case (e.g., when sampling by adding Gaussian noise), the known upperbounds based on log-concavity obtained for Gaussian mixtures by Saremi et al. (2024) is $\alpha \leqslant \frac{1}{d}$, which is significantly worse than our two-stage sampler.

## 5. Experiments

We now study how our new sampling scheme operates, first on synthetic data to understand the role of the noise parameter $\alpha$ and step-size $\eta$, then on binarized MNIST digits.

### 5.1. Synthetic data

We consider in this section mixtures of two independent binary vectors, that is, we consider, for $\beta > 0$,

$$p(x) = \frac{1}{(2\cosh\beta)^d}\left[\frac{1}{2}e^{\beta 1_n^\top x} + \frac{1}{2}e^{-\beta 1_n^\top x}\right],$$

for which all score functions can be computed (see Appendix F). When $\beta$ is small, $p$ is close to the uniform distribution (a "weak" prior), while when $\beta$ is large, $p$ is close to a sum of two Diracs at opposite points in the hypercube (a "strong" prior).

For $d$ small ($d = 8$ below), it is possible to perform all computations in closed form (e.g., with infinitely many replications), by computing transition matrices of size $2^d \times 2^d$. This allows to analyze precisely the denoising performance.

**Denoising performance with exact samples from $y$.** In Fig. 1, we consider three values of $\beta$ and vary $\alpha$. As expected, when $\alpha$ decreases, the noise increases, and the denoising performance degrades. When $\beta$ is large, the prior has a strong effect, so denoising helps. When $\beta$ is small, the prior is not strong, denoising has little effect. Note also that when $\alpha$ is large, denoising simply outputs $y$ (the threshold where it happens depend on the strength of the prior).

Moreover, since the upper bound in Lemma 2.3 is obtained from the mean-square-error, it shows that the denoising performance is significantly better than the bound suggests.

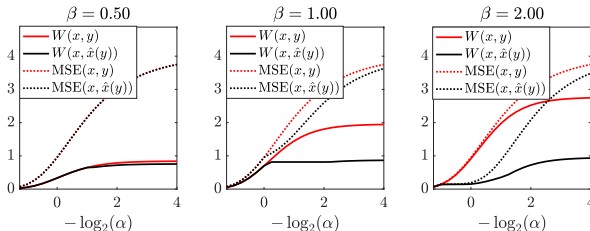

*Figure 1.* Optimal denoising from strong priors (large $\beta$) to weak priors (small $\beta$): comparison between Wasserstein distance and mean-square-error of denoising performance.

**Denoising performance with multiple measurements.**
In Fig. 2, we assess the benefits of multiple measurements by showing how the Wasserstein distance between our desired (noiseless) distribution on $x$ is estimated more closely by optimal denoising from $m$ measurements when $m$ is increasing, in particular for small $\alpha$ (high noise).

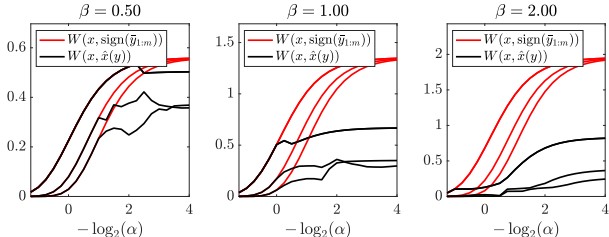

*Figure 2.* Optimal denoising from multiple measurements, for $m = 1, 3, 5$ (one curve per $m$), for $d = 6$, and three values of $\beta$, from strong priors (large $\beta$) to weak priors (small $\beta$).

**Comparisons of mixing times and distance to stationary distribution.** We compare our two sampling schemes (one-stage from Section 3.1, and two-stage from Section 3.2), and study the associated step sizes in terms of distance between the stationary distribution of the Markov chain and the desired distribution, and mixing time, which is here characterized by $1/(1 - \lambda_2)$, where $\lambda_2$ is the second largest eigenvalues of the transition matrix, a classical characterization of mixing time (Levin & Peres, 2017).

We see in Fig. 3 that when the step-size $\eta$ is too small, the mixing time explodes for all schemes (as predicted by Props. 3.1 and 3.3), and that for $\eta = 1/\alpha$ we obtain reasonable mixing times.

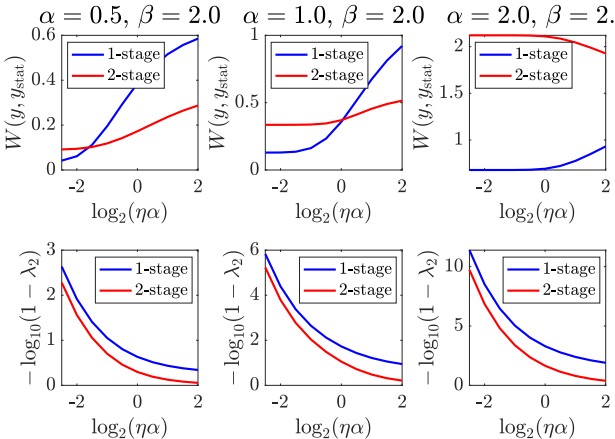

*Figure 3.* Comparison of 1-stage and 2-stage Langevin sampling. Top: distance to desired distribution $W(y, y_{\text{stat}})$, bottom: mixing time (in log scale).

**Denoising performance with samples obtained by discrete Langevin.** We consider in Fig. 4 a learning rate

equal to $1/\alpha$, and plot the Wasserstein distance to the distribution of $x$ for our two samplers. When $\alpha$ is large, the stationary distribution is far from the one of $y$, with bad performance. With $\alpha$ small, then the denoising performance is not great because too much noise is added. When $\beta$ is large (right plot), there is a clear sweet spot. Moreover, the two-stage sampler only provides improvements for small $\alpha$'s.

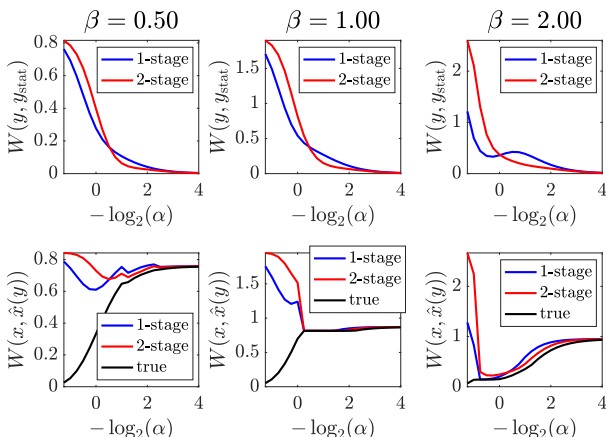

*Figure 4.* Comparison of 1-stage and 2-stage Langevin sampling. Top: distance to desired distribution $W(y, y_{\text{stat}})$, bottom: denoising performance of the stationary distributions, measured in Wasserstein distance.

### 5.2. Binarized MNIST

In this section we present our experiments on MNIST (LeCun et al., 1998). The clean binary data were prepared by scaling the pixel values be in $[0, 1]$ which we set as the probability of the Bernoulli distribution. The denoising is set up using logistic regression as outlined in Section 2.3, where we parametrize $f_\theta$ using the U-Net architecture (Ronneberger et al., 2015) with the modifications by Dhariwal & Nichol (2021). For optimization, we used AdamW (Loshchilov & Hutter, 2019) with the constant learning rate of $10^{-4}$ and the weight decay of $10^{-2}$. We present our experiments for $\alpha \in \{0.25, 0.5, 1, 2\}$ in Figs. 5 and 6.

**Denoising performance.** Fig. 5(a) shows 20 random binarized samples from the test set. Fig. 5 (b-d) shows the denoising performance of a trained model for very high noise, $\alpha = 0.25$, where we show both $\mathbb{E}[x|y]$, parametrized as $\tanh(f_\theta(y)/2)$ (see Section 2.3) and $\text{sign}(\mathbb{E}[x|y])$ which is optimal under the Hamming loss. In Fig. 5 (e-g) we repeat this experiment for a trained model at lower noise level $\alpha = 0.5$. Qualitatively, this is a sweet spot as the noise is high, yet the denoising performance is acceptable. Note the "5" flipping to "3", and "3" flipping to "8" by the denoiser due to the high noise. This already anticipates the fast mixing that could be achieved at this level of noise.

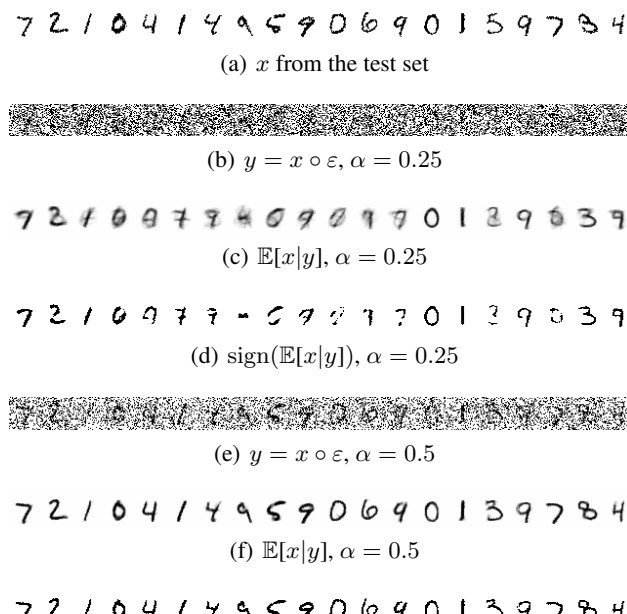

(a) $x$ from the test set

(b) $y = x \circ \varepsilon,\ \alpha = 0.25$

(c) $\mathbb{E}[x|y],\ \alpha = 0.25$

(d) $\mathrm{sign}(\mathbb{E}[x|y]),\ \alpha = 0.25$

(e) $y = x \circ \varepsilon,\ \alpha = 0.5$

(f) $\mathbb{E}[x|y],\ \alpha = 0.5$

(g) $\mathrm{sign}(\mathbb{E}[x|y]),\ \alpha = 0.5$

*Figure 5.* The denoising performance on binarized MNIST at two high Bernoulli noise levels ($\alpha = 0.25$, and $\alpha = 0.5$).

**Sampling performance.** Fig. 6 (a-f) illustrates the mixing performance of our algorithm for various noise levels. The step-size $\eta$ is set to $1/\alpha$ in all experiments. All panels show 100 steps of the algorithm, where the sampler is initialized at random bits. In Fig. 6 (a-c) we see the performance of the algorithm in "real time", where all the steps are shown ($\Delta k = 1$). These results show the remarkable mixing our algorithm can achieve. Fig. 6 (d-e) shows the typical performance of the algorithm for smaller noise $\alpha = 1$, where the samples are sharper but there is less mixing; here the results are shown by skipping 5 steps ($\Delta k = 5$). Finally, in Fig. 6(f) we see the sampling performance for smaller noise ($\alpha = 2$), where the sampling algorithm simply breaks down.

# 6. Conclusion

This study was motivated by whether we can reproduce the success of sampling through denoising while staying within the binary world. This required to reproduce the three key factors: (i) denoising through score functions, (ii) sampling noisy data via "smoothed" score functions, (iii) benefiting from multiple Bernoulli measurements. We achieved all three in an algorithmically simple framework, which comes with few hyperparameters (noise level $\alpha$ and number $m$ of measurements) and an arguably simpler formalism than discrete diffusions.

There are several avenues for future research: (1) our framework relies on using a noise process from an exponential family (here, Bernoulli) and can readily be extended to more

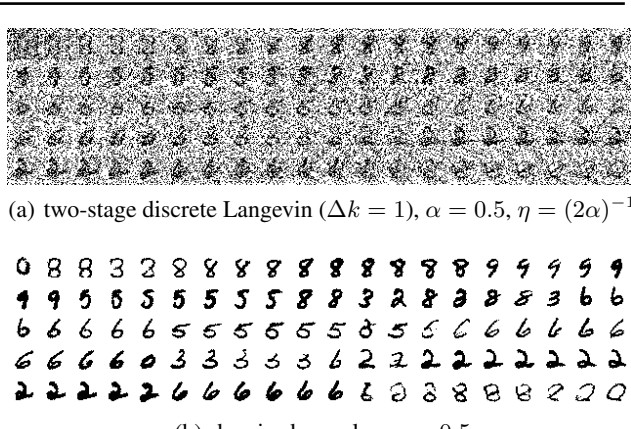

(a) two-stage discrete Langevin ($\Delta k = 1$), $\alpha = 0.5$, $\eta = (2\alpha)^{-1}$

(b) denoised samples, $\alpha = 0.5$

(c) single-stage discrete Langevin ($\Delta k = 1$), $\alpha = 0.5$, $\eta = 1/\alpha$

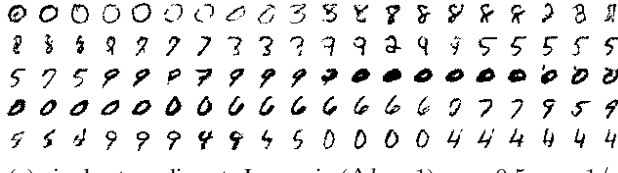

(d) single-stage discrete Langevin ($\Delta k = 5$), $\alpha = 1$

(e) single-stage discrete Langevin ($\Delta k = 5$), $\alpha = 1$, denoised

(f) single-stage discrete Langevin ($\Delta k = 5$), $\alpha = 2$

*Figure 6.* The sampling performance of our algorithm for binarized MNIST at three Bernoulli noise levels, visualized on single Markov chains (viewed left to right, top to bottom). (a) Two-stage discrete Langevin at $\alpha = 0.5$, (b) the denoised samples are shown, (c) due to space only denoised samples are shown for the vanilla (single-stage) algorithm, (d,e) here, $\alpha = 1$, and we skip every 5 steps, (f) $\alpha = 2$, denoised samples are not shown as the noise is small.

complex ones; (2) sharper denoising results for strong priors could also be examined; finally, (3) faster sampling could be achieved through the proper use of Metropolis-Hasting's step (Robert & Casella, 2004).

# Acknowledgements

We thank Armand Gissler, Alain Durmus, Dario Shariatian, and Eliot Beyler for discussions related to this work. This work has received support from the French government, managed by the National Research Agency, under the France 2030 program with the reference "PR[AI]RIE-PSAI" (ANR-23-IACL-0008).

## Impact Statement

This paper presents work whose goal is to advance the field of machine learning. There are many potential societal consequences of our work, none which we feel must be specifically highlighted here.

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

## A. Proof of Lemma 2.4

We denote $\varepsilon_1, \ldots, \varepsilon_m \in \{-1, 1\}^d$ the $m$ independent noise variables defined in Eq. (1). We assume that $m$ is odd for simplicity. The case $m$ even can be done similarly by splitting the case where $\sum_{i=1}^m \varepsilon_i = 0$.

Following the same reasoning that in the proof of Lemma 2.3, the Wasserstein distance is less than $d$ times the probability that $\sum_{i=1}^m (\varepsilon_i)_1 \leqslant 0$. Since $(\varepsilon_i)_1 = 2u_i - 1$ where $u_i$ is a Bernoulli random variable with parameter $\sigma(2\alpha) \in [1/2, 1]$. We need to upper bound, using the Chernoff bound,[1] the following probability as:

$$\mathbb{P}\Big(\frac{1}{m}\sum_{i=1}^m u_i \leqslant \frac{1}{2}\Big) \quad \leqslant \quad \exp(-m \cdot D\big(\tfrac{1}{2}\|\sigma(2\alpha)\big)),$$

where, for $\alpha \geqslant 0$, the Kullback-Leibler divergence between the uniform distribution and the Bernoulli distribution with parameter $\sigma(2\alpha)$ is equal to:

$$
\begin{aligned}
D\big(\tfrac{1}{2}\|\sigma(2\alpha)\big) &= \frac{1}{2}\log\frac{1}{2\sigma(2\alpha)} + \frac{1}{2}\log\frac{1}{2\sigma(-2\alpha)} \\
&= \frac{1}{2}\log\frac{1+e^{-2\alpha}}{2} + \frac{1}{2}\log\frac{1+e^{2\alpha}}{2} = \frac{1}{2}\log\frac{1+\cosh(2\alpha)}{2},
\end{aligned}
$$

which is always strictly greater than zero for $\alpha > 0$, equivalent to $\alpha - \log 2$ for large $\alpha$, and to $\frac{1}{2}\alpha^2$ for small $\alpha$.

## B. Proof of Proposition 3.1

*Proof.* We consider two random variables $y'$ and $z'$ marginally distributed from $t(\cdot|y)$ and $t(\cdot|z)$. We have, by definition of the Wasserstein distance:

$$
\begin{aligned}
W(t(\cdot|y), t(\cdot|z)) &= \inf_{\text{joint coupling}} \sum_{i=1}^d \mathbb{P}(y_i' \neq z_i') \text{ by definition of } W, \\
&\leqslant \sum_{i=1}^d \inf_{\text{marginal coupling}} \mathbb{P}(y_i' \neq z_i') \tag{11} \\
&\qquad \text{because we can construct canonically a joint coupling from marginal couplings,} \\
&= \sum_{i=1}^d \big|\mathbb{P}(y_i' = 1) - \mathbb{P}(z_i' = 1)\big| \text{ because of properties of total variation,} \\
&= \sum_{i=1}^d \Big|\sigma\big(\tfrac{2}{\eta}y_i + s(y)_i\big) - \sigma\big(\tfrac{2}{\eta}z_i + s(z)_i\big)\Big|, \tag{12}
\end{aligned}
$$

by definition of the transition kernel $t$ in Eq. (6). For proprieties of the total variation distance, see https://en.wikipedia.org/wiki/Total_variation_distance_of_probability_measures. We can then separate $i$'s according to $y_i = z_i$ or $y_i \neq z_i$, to get from Eq. (12):

$$
\begin{aligned}
W(t(\cdot|y), t(\cdot|z)) \quad \leqslant \quad & \sum_{i, y_i = z_i} \Big|\sigma\big(\tfrac{2}{\eta}y_i + s(y)_i\big) - \sigma\big(\tfrac{2}{\eta}z_i + s(z)_i\big)\Big| \\
& + \sum_{i, y_i = -z_i} \Big|\sigma\big(\tfrac{2}{\eta}y_i + s(y)_i\big) - \sigma\big(\tfrac{2}{\eta}z_i + s(z)_i\big)\Big|.
\end{aligned}
$$

---

[1] See https://en.wikipedia.org/wiki/Chernoff_bound.

We can now divide in two cases, whether $y_i = 1$ or $+1$, leading to

$$
\begin{aligned}
W(t(\cdot|y), t(\cdot|z)) \;\leqslant\; & \sum_{i, y_i = z_i = 1} \left| \sigma\left(\frac{2}{\eta} y_i + s(y)_i\right) - \sigma\left(\frac{2}{\eta} z_i + s(z)_i\right) \right| \\
& + \sum_{i, y_i = z_i = -1} \left| \sigma\left(\frac{2}{\eta} y_i + s(y)_i\right) - \sigma\left(\frac{2}{\eta} z_i + s(z)_i\right) \right| \\
& + \sum_{i, y_i = -z_i = 1} \left| \sigma\left(\frac{2}{\eta} y_i + s(y)_i\right) - \sigma\left(\frac{2}{\eta} z_i + s(z)_i\right) \right| \\
& + \sum_{i, y_i = -z_i = -1} \left| \sigma\left(\frac{2}{\eta} y_i + s(y)_i\right) - \sigma\left(\frac{2}{\eta} z_i + s(z)_i\right) \right| \\
=\; & \sum_{i, y_i = z_i = 1} \left| \sigma\left(\frac{2}{\eta} + s(y)_i\right) - \sigma\left(\frac{2}{\eta} + s(z)_i\right) \right| \\
& + \sum_{i, y_i = z_i = -1} \left| \sigma\left(-\frac{2}{\eta} + s(y)_i\right) - \sigma\left(-\frac{2}{\eta} + s(z)_i\right) \right| \\
& + \sum_{i, y_i = -z_i = 1} \left| \sigma\left(\frac{2}{\eta} + s(y)_i\right) - \sigma\left(-\frac{2}{\eta} + s(z)_i\right) \right| \\
& + \sum_{i, y_i = -z_i = -1} \left| \sigma\left(-\frac{2}{\eta} + s(y)_i\right) - \sigma\left(\frac{2}{\eta} + s(z)_i\right) \right|.
\end{aligned}
$$

We can now use the facts that $\sigma(-u) = 1 - \sigma(u)$, and that for $v, v' \geqslant u$, $\sigma'(v) = \sigma(v)\sigma(-v) = \frac{1}{2+e^{-v}+e^v} \leqslant \frac{1}{2+e^u}$, and thus, by Taylor's formula, $|\sigma(v) - \sigma(v')| \leqslant \frac{1}{2+e^u} |v - v'|$, to get

$$
\begin{aligned}
W(t(\cdot|y), t(\cdot|z)) \;\leqslant\; & \sum_{y_i = z_i} \frac{1}{2 + \exp\left(\frac{2}{\eta} - \beta_1\right)} \left| s(y)_i - s(z)_i \right| \\
& + \sum_i \mathbb{1}_{y_i \neq z_i} \left| \sigma\left(\frac{2}{\eta} + y_i s(y)_i\right) - \sigma\left(-\frac{2}{\eta} - y_i s(z)_i\right) \right|.
\end{aligned}
$$

We can now use the monotonicity of $\sigma$ and the bounds $\|s(y)\|_\infty, \|s(z)\|_\infty \leqslant \beta_1$, and the $\beta_2$-Lipschitz-continuity of $s$ (all from Eq. (5)) to get

$$
\begin{aligned}
W(t(\cdot|y), t(\cdot|z)) \;\leqslant\; & \sum_{y_i = z_i} \frac{1}{2 + \exp\left(\frac{2}{\eta} - \beta_1\right)} 2\beta_2 \ell(y, z) \\
& + \sum_i \mathbb{1}_{y_i \neq z_i} \left( \sigma\left(\frac{2}{\eta} + \beta_1\right) - \sigma\left(-\frac{2}{\eta} - \beta_1\right) \right) \\
\leqslant\; & \frac{1}{2 + \exp\left(\frac{2}{\eta} - \beta_1\right)} 2\beta_2 d\ell(y, z) + \left| 1 - 2\sigma\left(-\frac{2}{\eta} - \beta_1\right) \right| \cdot \ell(y, z) \\
\leqslant\; & \left[ \frac{2\beta_2 d}{2 + \exp\left(\frac{2}{\eta} - \beta_1\right)} + 1 - \frac{2}{1 + \exp\left(\frac{2}{\eta} + \beta_1\right)} \right] \cdot \ell(y, z) \\
\leqslant\; & \left[ 1 - \exp\left(-\frac{2}{\eta} - \beta_1\right) + 2\beta_2 d \exp\left(-\frac{2}{\eta} + \beta_1\right) \right] \cdot \ell(y, z).
\end{aligned}
$$

If $4\beta_2 d e^{2\beta_1} \leqslant 1$, then $2\beta_2 d \exp\left(-\frac{2}{\eta} + \beta_1\right) \leqslant \frac{1}{2} \exp\left(-\frac{2}{\eta} - \beta_1\right)$, and we get the desired result:

$$
W(t(\cdot|y), t(\cdot|z)) \leqslant \left(1 - \frac{1}{2} \exp\left(-\frac{2}{\eta} - \beta_1\right)\right) \ell(y, z).
$$

$\square$

## C. Proof of Proposition 3.2

*Proof.* For the purpose of the proof, we consider adding on top of the transition kernel $t$ a Metropolis-Hasting (MH) step (see, e.g., Robert & Casella, 2004) that keeps the proposal $y'$ given $y$ unchanged with probability

$$\min\left\{1, \frac{q(y')t(y|y')}{q(y)t(y'|y)}\right\}, \tag{13}$$

and go back to $y$ instead. The stationary distribution associated with this transition kernel is then exactly $q$.

We consider an arbitrary probability distribution $r$.

We consider two coupled samples $y$ from $r$ and $z$ from $q$, so that $W(r, q) = \mathbb{E}[\ell(y, z)]$. We also assume that, given $y, z$, the binary vectors $y', z'$ are sampled jointly respectively from $t(\cdot|y)$ and $t(\cdot|z)$, so that the Wasserstein distance given $y, z$ between the distributions $t(\cdot|y)$ and $t(\cdot|z)$ is $W(t(\cdot|y), t(\cdot|z)) = \mathbb{E}[\ell(y', z')|y, z]$. We consider $z''$ obtained from $z'$ by a Metropolis-Hasting step; $z''$ is then marginally distributed from $q$, while $y'$ is marginally distributed according to $\sum_{u \in \{-1,1\}^d} r(u)t(\cdot|u)$. Thus, by definition of $W$ as the loss for the optimal coupling, we have:

$$
\begin{aligned}
&W\Big(\textstyle\sum_{u \in \{-1,1\}^d} r(u)t(\cdot|u), q\Big) \\
&\leqslant \mathbb{E}[\ell(y', z'')] = \mathbb{E}[1_{\text{accept}(z,z')}\ell(z'', y')] + \mathbb{E}[1_{\text{reject}(z,z')}\ell(z'', y')] \\
&= \mathbb{E}[1_{\text{accept}(z,z')}\ell(z', y')] + \mathbb{E}[1_{\text{reject}(z,z')}\ell(z, y')] \text{ by definition of the MH step,} \\
&\leqslant \mathbb{E}[1_{\text{accept}(z,z')}\ell(z', y')] + \mathbb{E}[1_{\text{reject}(z,z')}(\ell(z', z) + \ell(z', y'))] \text{ by the triangular inequality,} \\
&= \mathbb{E}[\ell(z', y')] + \mathbb{E}[1_{\text{reject}(z,z')}\ell(z', z)] \\
&\leqslant \Big(1 - \frac{1}{2}\exp\big(-\frac{2}{\eta} - \beta_1\big)\Big)W(r, q) + \mathbb{E}[1_{\text{reject}(z,z')}\ell(z', z)],
\end{aligned}
$$

from the convergence result in Proposition 3.1. We have, by definition of the accept probability in Eq. (13),

$$
\begin{aligned}
\mathbb{E}[1_{\text{reject}(z,z')}\ell(z', z)] &= \sum_{z,z' \in \{-1,1\}^d} q(z)t(z'|z)\ell(z', z)\Big(1 - \min\Big\{1, \frac{q(z')t(z|z')}{q(z)t(z'|z)}\Big\}\Big) \tag{14} \\
&\leqslant \frac{1}{2}\sum_{z,z'} \ell(z', z)\big|q(z)t(z'|z) - q(z')t(z|z')\big|, \tag{15}
\end{aligned}
$$

with the transition kernel defined in Eq. (6), that is,

$$t(z'|z) \quad \propto \quad \exp\big((\tfrac{1}{2}s(z) + \tfrac{1}{\eta}z)^\top z'\big).$$

In order to prove a convergence result, we have to understand under which condition we obtain an approximate *detailed balance* condition (Levin & Peres, 2017). This will be a consequence of $s$ being small (e.g., here, $\nabla \log q(z)$ small).

We define the two additional transition kernels and distributions

$$
\begin{aligned}
\hat{t}(z'|z) &\propto \exp\big((\tfrac{1}{\eta}z)^\top z'\big) \\
\hat{q}(z) &\propto 1,
\end{aligned}
$$

for which we have the detailed balance condition $\hat{q}(z)\hat{t}(z'|z) - \hat{q}(z')\hat{t}(z|z') = 0$. We then get, from Eq. (15) and the triangular inequality,

$$\mathbb{E}[1_{\text{reject}(z,z')}\ell(z', z)] \leqslant \frac{1}{2}\sum_{z,z'} \ell(z', z)\big|q(z) - \hat{q}(z)\big| \cdot \hat{t}(z'|z) + \frac{1}{2}\sum_{z,z'} \ell(z', z)q(z) \cdot \big|\hat{t}(z'|z) - t(z'|z)\big|,$$

using detailed balance for $\hat{q}$ and $\hat{t}$. For the left term, we can use the symmetry of $\hat{t}$ and an explicit computation, to get

$$
\begin{aligned}
\sum_{z,z'} \ell(z',z)|q(z) - \hat{q}(z)| \cdot \hat{t}(z'|z) &= \sum_z |q(z) - \hat{q}(z)| \cdot \sum_{z'} \hat{t}(z'|1_d)\ell(z',1_d) \\
&= \sum_z |q(z) - \hat{q}(z)| \cdot \sum_{i=1}^d \hat{t}(z'_i = -1|1_d) \text{ since } \hat{t}(\cdot|1_d) \text{ has independent components,} \\
&\leqslant 2d\sigma(-2/\eta)\mathrm{TV}(q,\hat{q}) \leqslant 2d\exp(-2/\eta)\mathrm{TV}(q,\hat{q}),
\end{aligned}
$$

where $\mathrm{TV}(q,\hat{q}) = \frac{1}{2}\sum_z |q(z) - \hat{q}(z)|$ is the total variation distance[2] between $q$ and $\hat{q}$. For the second term, we have

$$
\begin{aligned}
\sum_{z,z'} \ell(z',z)q(z) \cdot |\hat{t}(z'|z) - t(z'|z)| &\leqslant \max_z \sum_{z'} \ell(z',z)|\hat{t}(z'|z) - t(z'|z)| \\
&\leqslant d \cdot \max_z \sum_{z'} |\hat{t}(z'|z) - t(z'|z)| \text{ since } \ell(\cdot,\cdot) \leqslant d, \\
&\leqslant d \cdot \frac{4d\beta_1}{2 + e^{2/\eta - \beta_1}}.
\end{aligned}
$$

We have used the small lemma for the two probability mass functions on $\{-1,1\}^d$ proportional to $A(y) \propto e^{y^\top a}$ and $B(y) \propto e^{y^\top b}$: $\sum_y |A(y) - B(y)| \leqslant 2\sum_{i=1}^d |\sigma(2a_i) - \sigma(2b_i)| \leqslant 2\sum_{i=1}^d \frac{2}{2 + \exp(\min\{2a_i, 2b_i\})}|a_i - b_i|$.

Overall, we get

$$
W\Big(\sum_{u\in\{-1,1\}^d} r(u)t(\cdot|u), q\Big) \leqslant \Big(1 - \frac{1}{2}\exp\Big(-\frac{2}{\eta} - \beta_1\Big)\Big)W(r,q) + d\exp(-2/\eta)\mathrm{TV}(q,\hat{q}) + d \cdot \frac{2d\beta_1}{e^{2/\eta - \beta_1}}.
$$

Thus, if $q'$ denotes the stationary distribution of the Markov kernel $t$, we get, applying the above inequality to $r = q'$,

$$
W(q',q) \leqslant \Big(1 - \frac{1}{2}\exp\Big(-\frac{2}{\eta} - \beta_1\Big)\Big)W(q',q) + d\exp(-2/\eta)\mathrm{TV}(q,\hat{q}) + d \cdot \frac{2d\beta_1}{e^{2/\eta - \beta_1}},
$$

leading to

$$
W(q',q) \leqslant 2de^{\beta_1}\mathrm{TV}(q,\hat{q}) + 4d^2\beta_1 e^{2\beta_1}.
$$

We can now use Pinsker's inequality[3], to get:

$$
\begin{aligned}
\mathrm{TV}(q,\hat{q}) &\leqslant \Big(\frac{1}{2}\mathrm{KL}(q\|\hat{q})\Big)^{1/2} = \Big(\frac{1}{2}\mathbb{E}_{q(z)}[\log q(z) - \log \frac{1}{2^d}]\Big)^{1/2} \\
&\leqslant \Big(\frac{1}{2}\mathbb{E}_{q(z)}[\log q(z_0) + \beta_1\|z - z_0\|_1 - \log \frac{1}{2^d}]\Big)^{1/2} \text{ using the boundedness of } s, \\
&\leqslant \Big(\frac{1}{2}\mathbb{E}_{q(z)}[\beta_1\|z - z_0\|_1]\Big)^{1/2} \leqslant \sqrt{\beta_1 d},
\end{aligned}
$$

by choosing $z_0$ such that $q(z_0) \leqslant \frac{1}{2^d}$ (there has to be one). We thus get the desired result

$$
W(q',q) \leqslant 2de^{\beta_1}\sqrt{\beta_1 d} + 4d^2\beta_1 e^{2\beta_1} = 2d\big(2d\beta_1 e^{2\beta_1} + \sqrt{d\beta_1 e^{2\beta_1}}\big).
$$

$\square$

# D. Proof of Proposition 3.3

*Proof.* We can reuse the proof for the one-stage sampler to obtain the contractivity of the second step of the two-stage sampler (updating the fact that we have no $1/2$ factor) to get a contracting rate

$$
\Big[1 - \exp\Big(-\frac{2}{\eta} - 2\beta_1\Big) + 4\beta_2 d\exp\Big(-\frac{2}{\eta} + 2\beta_1\Big)\Big].
$$

---

[2]See https://en.wikipedia.org/wiki/Total_variation_distance_of_probability_measures.
[3]See https://en.wikipedia.org/wiki/Pinsker%27s_inequality.

We thus need the condition $8\beta_2 d e^{4\beta_1} \leqslant 1$ to get the contraction

$$\Big(1 - \frac{1}{2}\exp\big(-\frac{2}{\eta} - 2\beta_1\big)\Big).$$

We now need to compute the contraction for the first step using simply $\beta_1 = \beta_2 = 0$ in the same reasoning, leading to a contraction

$$\Big(1 - \exp\big(-\frac{2}{\eta}\big)\Big).$$

Multiplying the two contractions, we get

$$\Big(1 - \frac{1}{2}\exp\big(-\frac{2}{\eta} - 2\beta_1\big)\Big)\Big(1 - \exp\big(-\frac{2}{\eta}\big)\Big) \leqslant \Big(1 - \frac{1}{2}\exp\big(-\frac{2}{\eta} - 2\beta_1\big)\Big),$$

which leads to the desired result. $\qquad\qquad\square$

## E. Proof of Proposition 3.4

*Proof.* We consider $y^{(1)} \in \{-1, 1\}^d$ distributed from an arbitrary distribution $r$, and $y^{(2)}$ sampled from the distribution $q$. We sample $z^{(1)}$ and $z^{(2)}$ from $u(\cdot|y^{(1)})$ and $u(\cdot|y^{(2)})$, as well as $\bar{y}^{(1)}$ and $\bar{y}^{(2)}$, from $u(\cdot|z^{(1)})$ and $u(\cdot|z^{(2)})$ (so that we get a full approximate Gibbs sampling step, with transition kernel $v$, from $y^{(1)}$ to $\bar{y}^{(1)}$ and $y^{(2)}$ to $\bar{y}^{(2)}$), all coupled so that the Wasserstein distance between

$$\sum_{y^{(1)}} r(y^{(1)})v(\cdot|y^{(1)})$$

and

$$\sum_{y^{(2)}} q(y^{(2)})v(\cdot|y^{(2)})$$

(that is, one step of the Markov transition kernel) is less than $\mathbb{E}[\ell(\bar{y}^{(1)}, \bar{y}^{(2)})]$.

For the purpose of the proof, we can now add a Metropolis Hasting step to the second chain (which leads to $\bar{\bar{y}}^{(2)}$) so that, since it starts from the stationary distribution $q$ of the full Gibbs sampling step, it remains at $q$. Thus, like in Appendix C,

$$
\begin{aligned}
W\Big(\sum_{y^{(1)}} r(y^{(1)})v(\cdot|y^{(1)}), q\Big) \quad &\leqslant \quad \mathbb{E}[\ell(\bar{y}^{(1)}, \bar{\bar{y}}^{(2)})] \\
&\leqslant \quad \mathbb{E}[\ell(\bar{y}^{(1)}, \bar{y}^{(2)})] + \mathbb{E}[\ell(\bar{y}^{(2)}, \bar{\bar{y}}^{(2)})] \\
&\leqslant \quad \Big(1 - \frac{1}{2}\exp\big(-\frac{2}{\eta} - 2\beta_1\big)\Big)W(r, q) + \mathbb{E}[1_{\mathrm{reject}}(y^{(2)}, \bar{y}^{(2)})\ell(\bar{y}^{(2)}, y^{(2)})].
\end{aligned}
$$

Moreover, like in Appendix C, we have, now dropping the superscripts $^{(2)}$:

$$\mathbb{E}[1_{\mathrm{reject}}(y, \bar{y})\ell(\bar{y}, y)] \quad = \quad \frac{1}{2}\sum_{y, \bar{y}} \ell(\bar{y}, y)\big|q(y)v(\bar{y}|y) - q(\bar{y})v(y|\bar{y})\big|.$$

We have, using that $q(z|y) = u(z|y)$, and by definition of $v$,

$$q(y)v(\bar{y}|y) = \sum_z q(y)u(\bar{y}|z)u(z|y) = \sum_z q(y)u(\bar{y}|z)q(z|y) = \sum_z q(z)q(y|z)u(\bar{y}|z),$$

because $q(y)q(z|y) = q(y,z) = q(z)q(y|z)$, leading to:

$$
\begin{aligned}
\mathbb{E}[1_{\text{reject}}(y,\bar{y})\ell(\bar{y},y)] &= \frac{1}{2}\sum_{y,\bar{y}} \ell(\bar{y},y)\left|\sum_z q(z)q(y|z)u(\bar{y}|z) - \sum_z q(z)q(\bar{y}|z)u(y|z)\right| \\
&= \frac{1}{2}\sum_{y,\bar{y}} \ell(\bar{y},y)\left|\sum_z q(z)\big(q(y|z)u(\bar{y}|z) - q(\bar{y}|z)u(y|z)\big)\right| \\
&\leqslant \frac{1}{2}\sum_{z,y,\bar{y}} \ell(\bar{y},y)q(z)\big|q(y|z)u(\bar{y}|z) - u(y|z)q(\bar{y}|z)\big| \text{ by the triangular inequality,} \\
&\leqslant \frac{1}{2}\max_z \sum_{y,\bar{y}} \ell(\bar{y},y)|q(y|z)u(\bar{y}|z) - u(y|z)q(\bar{y}|z)| \text{ by bounding the expectation by the max,} \\
&= \frac{1}{2}\max_z \sum_{y,\bar{y}} \ell(\bar{y},y)|q(y|z)u(\bar{y}|z) - u(y|z)u(\bar{y}|z) + u(y|z)u(\bar{y}|z) - u(y|z)q(\bar{y}|z)| \\
&\leqslant \frac{1}{2}\max_z \sum_{y,\bar{y}} \ell(\bar{y},y)\Big\{u(\bar{y}|z)\big|q(y|z) - u(y|z)\big| + u(y|z)\big|u(\bar{y}|z) - q(\bar{y}|z)\big|\Big\} \\
&= \max_z \sum_{y,\bar{y}} \ell(\bar{y},y)u(y|z)|q(\bar{y}|z) - u(\bar{y}|z)| \text{ by symmetry,} \\
&\leqslant \max_z \sum_{y,\bar{y}} \big[\ell(\bar{y},z) + \ell(z,y)\big]u(y|z)|q(\bar{y}|z) - u(\bar{y}|z)| \text{ by the triangular inequality,} \\
&\leqslant \max_z \left\{\sum_{y,\bar{y}} \ell(\bar{y},z)u(y|z)|q(\bar{y}|z) - u(\bar{y}|z)| + \sum_{y,\bar{y}} \ell(z,y)u(y|z)|q(\bar{y}|z) - u(\bar{y}|z)|\right\} \\
&\qquad\qquad\qquad\qquad\qquad\qquad\qquad\qquad \text{by separating the sum,} \\
&= \max_z \left\{\sum_{\bar{y}} \ell(\bar{y},z)|q(\bar{y}|z) - u(\bar{y}|z)| + \sum_y \ell(z,y)u(y|z)\sum_{\bar{y}} |q(\bar{y}|z) - u(\bar{y}|z)|\right\} \\
&\qquad\qquad\qquad\qquad\qquad\qquad\qquad\qquad \text{by summing out } y \text{ in the first term,} \\
&= \max_z \sum_{\bar{y}} \left[\ell(\bar{y},z) + d\sigma(-\frac{2}{\eta} + 2\beta_1)\right]\cdot|q(\bar{y}|z) - u(\bar{y}|z)|, \\
&\leqslant \max_z \sum_{\bar{y}} \ell(\bar{y},z)\cdot|q(\bar{y}|z) - u(\bar{y}|z)| + 2d\sigma(-\frac{2}{\eta} + 2\beta_1)\max_z \text{TV}(q(\cdot|z), u(\cdot|z)), \quad (16)
\end{aligned}
$$

using that $\sum_y \ell(z,y)u(y|z) = \sum_{i=1}^{d} \mathbb{P}_{u(y_i|z_i)}(y_i \neq z_i|z_i) = \sum_{i=1}^{d} \sigma(-2/\eta - 2s(z)_i) \leqslant d\sigma(-2/\eta + 2\beta_1)$ and that $u(\cdot|z)$ has independent components.

We can now write, because of our assumption in Eq. (10),

$$
q(y|z) = u(y|z)\frac{1}{Z(z)}e^{\varphi(y,z)},
$$

with $\varphi(y,z) = \log q(y) - \log q(z) - s(z)^\top(y-z)$, which satisfies

$$
0 \leqslant \varphi(y,z) \leqslant \frac{\beta_2}{2}\|y-z\|_1^2,
$$

and

$$
\begin{aligned}
\log Z(z) &= \log \sum_y u(y|z) e^{\varphi(y,z)} \leqslant \log \sum_y u(y|z) e^{\frac{\beta_2}{2}\|y-z\|_1^2} \\
&\leqslant \log \sum_y u(y|z) e^{d\beta_2\|y-z\|_1} = \sum_{i=1}^d \log \sum_{y_i} u(y_i|z_i) e^{d\beta_2|y_i - z_1|} \text{ using that } \|y-z\|_1 \leqslant 2d, \\
&\leqslant \sum_{i=1}^d \log \Big( \sigma(2/\eta + 2s(z)_i) + \sigma(-2/\eta - 2s(z)_i) e^{2d\beta_2} \Big) \text{ by definition of } u(\cdot|z), \\
&= \sum_{i=1}^d \log \Big( 1 + \sigma(-2/\eta - 2s(z)_i)(e^{2d\beta_2} - 1) \Big) \leqslant d \log \Big( 1 + \sigma(-2/\eta + 2\beta_1)(e^{2d\beta_2} - 1) \Big) \\
&\leqslant d \log \Big( 1 + e^{-2/\eta + 2\beta_1}(e^{2d\beta_2} - 1) \Big) \leqslant d e^{-2/\eta + 2\beta_1}(e^{2d\beta_2} - 1) \text{ using } \log(1+c) \leqslant c. \quad (17)
\end{aligned}
$$

Moreover, we have $Z(z) \geqslant 1$.

We treat the two terms in Eq. (16) separately. For the second term, we have, using Pinsker's inequality,

$$
\begin{aligned}
\mathrm{TV}(q(\cdot|z), u(\cdot|z)) &\leqslant \Big( \frac{1}{2} \mathrm{KL}(u(\cdot|z)\|q(\cdot|z)) \Big)^{1/2} = \Big( \frac{1}{2} \mathbb{E}_{u(y|z)} \log \frac{u(y|z)}{q(y|z)} \Big)^{1/2} \\
&\leqslant \Big( \frac{1}{2} \mathbb{E}_{u(y|z)} \log Z(z) - \varphi(y,z) \Big)^{1/2} \leqslant \Big( \frac{1}{2} \log Z(z) \Big)^{1/2},
\end{aligned}
$$

because $\varphi \geqslant 0$. Thus the second term in Eq. (16) can be bounded as follows, using Eq. (17):

$$
\begin{aligned}
2d\sigma(-\frac{2}{\eta} + 2\beta_1) \max_z \mathrm{TV}(q(\cdot|z), u(\cdot|z)) &\leqslant 2d e^{-\frac{2}{\eta} + 2\beta_1} \Big( \frac{d}{2} e^{-2/\eta + 2\beta_1}(e^{2d\beta_2} - 1) \Big)^{1/2} \\
&= \sqrt{2} d^{3/2} e^{-\frac{3}{\eta} + 3\beta_1}(e^{2d\beta_2} - 1)^{1/2}. \quad (18)
\end{aligned}
$$

For the first term in Eq. (16), we have:

$$
\begin{aligned}
\sum_{\bar{y}} \ell(\bar{y}, z) \cdot \big| q(\bar{y}|z) - u(\bar{y}|z) \big| &= \sum_{\bar{y}} \sum_{i=1}^d 1_{\bar{y}_i \neq z_i} u(\bar{y}|z) \cdot \Big| \frac{1}{Z(z)} e^{\varphi(\bar{y},z)} - 1 \Big| \text{ by definition of } \varphi, \\
&= \sum_{\bar{y}} \sum_{i=1}^d 1_{\bar{y}_i \neq z_i} \sigma(-2/\eta - 2s(z)_i) \prod_{j \neq i} u(\bar{y}_j|z_j) \cdot \Big| \frac{1}{Z(z)} e^{\varphi(\bar{y},z)} - 1 \Big| \text{ by definition of } u.
\end{aligned}
$$

We now use the inequality

$$
\begin{aligned}
\Big| \frac{1}{Z(z)} e^{\varphi(\bar{y},z)} - 1 \Big| &= \Big( \frac{1}{Z(z)} e^{\varphi(\bar{y},z)} - 1 \Big)_+ + \Big( 1 - e^{\varphi(\bar{y},z) - \log Z(z)} \Big)_+ \\
&\leqslant \Big( e^{\varphi(\bar{y},z)} - 1 \Big)_+ + \Big( \log Z(z) - \varphi(\bar{y},z) \Big)_+ \leqslant e^{\varphi(\bar{y},z)} - 1 + \log Z(z),
\end{aligned}
$$

which is the result of $\varphi \geqslant 0$ and $Z \geqslant 1$, to get, using $\varphi(\bar{y}, z) \leqslant d\beta_2 \|\bar{y} - z\|_1$,

$$\sum_{\bar{y}} \ell(\bar{y}, z) \cdot \big| q(\bar{y}|z) - u(\bar{y}|z) \big|$$

$$\leqslant \sum_{\bar{y}} \sum_{i=1}^{d} 1_{\bar{y}_i \neq z_i} \sigma(-2/\eta - 2s(z)_i) \prod_{j \neq i} u(\bar{y}_j|z_j) \cdot \big| e^{d\beta_2 \|\bar{y} - z\|_1} - 1 + \log Z(z) \big|$$

$$\leqslant \sum_{i=1}^{d} \sum_{\bar{y}_j, j \neq i} \sigma(-2/\eta + 2\beta_1) \prod_{j \neq i} u(\bar{y}_j|z_j) \cdot \Big( e^{2d\beta_2} e^{d\beta_2 \sum_{j \neq i} |\bar{y}_j - z_j|} - 1 + \log Z(z) \Big)$$

$$= \sigma(-2/\eta + 2\beta_1) \sum_{i=1}^{d} \Big( e^{2d\beta_2} \prod_{j \neq i} \big\{ \sigma(2/\eta + 2s(z)_j) + \sigma(-2/\eta - 2s(z)_j) e^{2d\beta_2} \big\} - 1 + \log Z(z) \Big)$$

$$= \sigma(-2/\eta + 2\beta_1) \sum_{i=1}^{d} \Big( e^{2d\beta_2} \prod_{j \neq i} \big\{ 1 + \sigma(-2/\eta + 2s(z)_j)(e^{2d\beta_2} - 1) \big\} - 1 + \log Z(z) \Big)$$

$$\leqslant e^{-2/\eta + 2\beta_1} \sum_{i=1}^{d} \Big( e^{2d\beta_2} \prod_{j \neq i} \big\{ 1 + \sigma(-2/\eta + 2\beta_1)(e^{2d\beta_2} - 1) \big\} - 1 + \log Z(z) \Big)$$

$$= d e^{-2/\eta + 2\beta_1} \Big( e^{2d\beta_2} \big( 1 + \sigma(-2/\eta + 2\beta_1)(e^{2d\beta_2} - 1) \big)^{d-1} - 1 + \log Z(z) \Big).$$

From the constraint $8d\beta_2 e^{4\beta_1} \leqslant 1$, we have $\beta_2 d \leqslant \frac{1}{8}$, which implies that $e^{2d\beta_2} - 1 \leqslant \frac{5}{2}d\beta_2$. Moreover, we assume that $e^{-2/\eta + 2\beta_1} \leqslant \frac{1}{d}$. This leads to, using Eq. (17),

$$\sum_{\bar{y}} \ell(\bar{y}, z) \cdot \big| q(\bar{y}|z) - u(\bar{y}|z) \big| \;\leqslant\; d e^{-2/\eta + 2\beta_1} \Big( e^{2d\beta_2} \big( 1 + \frac{1}{d}\frac{5}{2}d\beta_2 \big)^d - 1 + d e^{-2/\eta + 2\beta_1}(e^{2d\beta_2} - 1) \Big)$$

$$\leqslant\; d e^{-2/\eta + 2\beta_1} \Big( e^{2d\beta_2} e^{\frac{5}{2}d\beta_2} - 1 + d e^{-2/\eta + 2\beta_1} \frac{5}{2}d\beta_2 \Big)$$

$$\leqslant\; d e^{-2/\eta + 2\beta_1} 6d\beta_2 + \frac{5}{2}d^3\beta_2 e^{-4/\eta + 4\beta_1}, \tag{19}$$

using $1 + c \leqslant e^c$.

Thus, assembling the terms in Eq. (18) and Eq. (19),

$$\mathbb{E}[1_{\text{reject}}(y, \bar{y})\ell(\bar{y}, y)] \;\leqslant\; d e^{-2/\eta + 2\beta_1} 6d\beta_2 + \frac{5}{2}d^3\beta_2 e^{-4/\eta + 4\beta_1} + \sqrt{2}d^{3/2} e^{-\frac{3}{\eta} + 3\beta_1} \big( \frac{5}{2}d\beta_2 \big)^{1/2}.$$

Thus, using the same reasoning as in Appendix C, and using that $\beta_2 d \leqslant \frac{1}{8}$,

$$W(q', q) \;\leqslant\; 2e^{\frac{2}{\eta} + 2\beta_1} \mathbb{E}[1_{\text{reject}}(y, \bar{y})\ell(\bar{y}, y)]$$

$$\leqslant\; 2d e^{4\beta_1} 6d\beta_2 + 5d^3\beta_2 e^{-2/\eta + 6\beta_1} + 2\sqrt{2}d^{3/2} e^{-\frac{1}{\eta} + 5\beta_1} \big( \frac{5}{2}d\beta_2 \big)^{1/2}$$

$$\leqslant\; 17d^2 e^{4\beta_1}\beta_2 + 2\sqrt{5}d^2 e^{-\frac{1}{\eta} + 5\beta_1} \sqrt{\beta_2} \text{ using } e^{-2/\eta + 2\beta_1} \leqslant \frac{1}{d}, \tag{20}$$

$$=\; 17d^2 e^{4\beta_1}\beta_2 + 2\sqrt{5}d^2 e^{-\frac{1}{\eta} + \beta_1} e^{4\beta_1} \sqrt{\beta_2}$$

$$\leqslant\; 17d^2 e^{4\beta_1}\beta_2 + 2\sqrt{5}d^2 \frac{1}{\sqrt{d}} e^{4\beta_1} \sqrt{\beta_2} = d e^{4\beta_1}[17d\beta_2 + \sqrt{20d\beta_2}] \leqslant d e^{4\beta_1}[17\frac{1}{\sqrt{8}}\sqrt{d\beta_2} + \sqrt{20d\beta_2}]$$

$$\leqslant\; 12d e^{4\beta_1}\sqrt{d\beta_2}.$$

Overall, we get a bound in $d$ times $\sqrt{\beta_2 d^2}$ and $d$ times $\beta_2 d^2$ if $\eta$ is small enough, by Eq. (20).

$\square$

# F. Mixtures of independent variables

In this section, we provide details on score functions for mixtures of two independent variables. We start with a few facts about independent variables.

**A few facts about independent variables.** If $p(x) \propto e^{\beta^\top x}$, then

$$p(x) = \frac{e^{\beta^\top x}}{\prod_{i=1}^{d} 2 \cosh \beta_i}$$

and

$$\sum_{x \in \{-1,1\}^d} p(x) x = \tanh(\beta x)$$

(taken componentwise).

If $q(y) \propto \sum_{x \in \{-1,1\}^d} p(x) e^{\alpha y^\top x}$, then $y = x \circ z$, where $p(z) \propto e^{\alpha 1_n^\top x}$ is independent from $x$. Since for independent variables, the first moment characterizes the distributions, we have

$$q(y) \propto e^{\gamma^\top x},$$

with $\tanh \gamma_i = \tanh \alpha \cdot \tanh \beta_i$. We then have two different formulas for $q(y)$:

$$\begin{aligned}
q(y) &= \frac{e^{\gamma^\top x}}{\prod_{i=1}^{d} 2 \cosh \gamma_i} \\
&= \frac{1}{(2 \cosh \alpha)^d} \frac{\prod_{i=1}^{d} \cosh(\beta_i + \alpha y_i)}{\prod_{i=1}^{d} \cosh \beta_i},
\end{aligned}$$

the second being obtained by computing the sum with respect to $x$. Note that these two formulas are equal for $y \in \{-1, 1\}^d$, not for generic $y$'s. Moreover, one can check that $\alpha = 0$ or $\beta = 0$ lead to a constant $q$.

We can compute $\mathbb{E}[x|y]$ as

$$\mathbb{E}[x|y] = \frac{1}{\alpha} \nabla \log q(x).$$

Using the first formula for $q$ leads to $\mathbb{E}[X|Y = y] = \frac{\gamma}{\alpha}$, which is incorrect. With the second formula, we get:

$$\frac{1}{\alpha} \nabla \log q(y) = \tanh(\beta + \alpha y).$$

**Mixtures of two independent variables.** We consider

$$p(x) = \frac{1}{(2 \cosh \beta)^d} \left[ \frac{1}{2} e^{\beta 1_n^\top x} + \frac{1}{2} e^{-\beta 1_n^\top x} \right].$$

With $\tanh \gamma = \tanh \alpha \cdot \tanh \beta$, then $q$ is a mixture of $\frac{1}{(2 \cosh \gamma)^d} e^{\gamma 1_n^\top y}$ and $\frac{1}{(2 \cosh \gamma)^d} e^{-\gamma 1_n^\top y}$, with

$$q(y) = \frac{1}{(2 \cosh \alpha)^d (\cosh \beta)^d} \left[ \frac{1}{2} \prod_{i=1}^{d} \cosh(\beta + \alpha y_i) + \frac{1}{2} \prod_{i=1}^{d} \cosh(\beta - \alpha y_i) \right],$$

and

$$\begin{aligned}
\frac{1}{\alpha} \nabla \log q(y) &= \frac{\prod_{i=1}^{d} \cosh(\beta + \alpha y_i) \cdot \tanh(\beta + \alpha y) - \prod_{i=1}^{d} \cosh(\beta - \alpha y_i) \cdot \tanh(\beta - \alpha y)}{\prod_{i=1}^{d} \cosh(\beta + \alpha y_i) + \prod_{i=1}^{d} \cosh(\beta - \alpha y_i)} \\
&= \frac{e^{\gamma 1_n^\top x} \tanh(\beta + \alpha y) - e^{-\gamma 1_n^\top x} \tanh(\beta - \alpha y)}{e^{\gamma 1_n^\top x} + e^{-\gamma 1_n^\top x}}.
\end{aligned}$$

Above, the first formula is valid for all $y \in \mathbb{R}^d$, while the second formula is only true for $y \in \{-1, 1\}^d$.

