# OpenReview forum: "Sampling Binary Data by Denoising through Score Functions"
_ICML.cc/2025/Conference — ICML 2025 poster_

### Official Review · Reviewer_REiK · 2025-03-11

**Overall Recommendation:** 2

**Summary:**

This paper proposes a generative model for data on the boolean hypercube. The basic idea is to "noise" the data with random sign flips, with expected number of sign flips controlled by a parameter $\alpha$, and then learn to "reverse" this operation which ultimately boils down to learning a denoiser in this paper. The denoiser takes as input a sample from the sign-flipped data distribution which is denoted by  $q_\alpha$. The overall sampling procedure proceeds as follows: first sample $Y \sim q_\alpha$ then query the denoiser at $Y$.

This sampling procedure introduces a specific compromise: as $\alpha$ is close to $0$, $q_\alpha$ becomes closer to the uniform distribution and so is easier to sample but then the law of ${denoiser}_{\alpha} (Y)$ can be very different from the target distribution.

A workaround is to include multiple measurements i.i.d. $Y_1, ..., Y_m$. In this case the denoiser takes as input these measurements altogether and its denoising performance, as measured by the closeness of the law of $denoiser_\alpha(Y_1, ..., Y_m)$, improves. Having trained the denoiser, sampling is performed by drawing approximately from the joint marginal distribution of $Y_1, ..., Y_m$. This is done by done first sampling $Y_1 \sim q_\alpha$ and then consecutive conditionals $Y_k | Y_{1:k-1}$. Crucially, we can take $\alpha$ to be small in this case so as to ensure that the distribution is easy to sample while at the same time having improved denoiser performance due to the multi-measurements. Since we are dealing with a discrete space, the authors proceed to develop a gibbs sampler for the initial distribution.

**Claims And Evidence:**

The paper makes a few interesting theoretical claims about the method which helps the reader gain intuition about how the proposed sampler behaves. In terms of evidence, most of them are addressed with theoretical results. One point which I believed is not substantiated with some evidence is the sampling of the conditional distributions. It seems that this is not discussed and is assumed to be rather easy to solve?

**Essential References Not Discussed:**

While not essential, the authors should also discuss the recent works [1, 2].

[1] Sahoo, S., Arriola, M., Schiff, Y., Gokaslan, A., Marroquin, E., Chiu, J., Rush, A. and Kuleshov, V., 2024. Simple and effective masked diffusion language models. Advances in Neural Information Processing Systems, 37, pp.130136-130184.

[2] Shi, J., Han, K., Wang, Z., Doucet, A. and Titsias, M., 2024. Simplified and generalized masked diffusion for discrete data. Advances in neural information processing systems, 37, pp.103131-103167.

**Experimental Designs Or Analyses:**

The experiments are sound; there is a toy experiment in which everything can be computed in closed form. It allows the reader to get a good grasp of how the various hyper parameters in the paper influence the final performance. The qualitative analysis of this experiment is interesting. The algorithm is then tested on a simple image experiment. Here again the qualitative analysis of the hyperparameters is sound and interesting.

It seems to me that this section lacks a comparison with existing methods that showcases its appeal.

**Methods And Evaluation Criteria:**

In my opinion the methodology lacks motivation. For example, the popular denoising diffusion models proceed by reversing a forward Markov chain that turns the data into noise. The initial distribution of this reverse process can be sampled almost exactly and then each transition in the backwad process can be sampled quite accurately as long as we have a well-trained denoiser.

This paper takes an orthogonal approach by instead relying on multiple measurements that are at the same "noise level". Due to this design choice, the initial distribution is not trivial to sample from and requires quite a machinery. The authors derive (partial) theoretical guarantees for the sampler they use for this initial distribution but still, the distance of the stationary distribution of their sampler to that of the target is not necessarily small. Furthermore, I believe that no guarantees are given for the conditional distributions that need to be sampled afterwards.

Essentially, I do not think that the paper gives compelling arguments as to why this is needed and what is its advantage compared to more traditional approaches. For example one could think about designing a forward Makov chain of sign flips, and this would converge to the uniform distribution on the hypercube exponentially fast. This Markov chain can then be reversed and having learned the "denoiser", I believe that the sampling could be done straightforwardly. This is essentially the analog of the approaches considered in the Masked Diffusion literature [1]. I would really like to understand in which cases this approach is more appealing.

[1] Sahoo, S., Arriola, M., Schiff, Y., Gokaslan, A., Marroquin, E., Chiu, J., Rush, A. and Kuleshov, V., 2024. Simple and effective masked diffusion language models. Advances in Neural Information Processing Systems, 37, pp.130136-130184.

[2] Shi, J., Han, K., Wang, Z., Doucet, A. and Titsias, M., 2024. Simplified and generalized masked diffusion for discrete data. Advances in neural information processing systems, 37, pp.103131-103167.

**Other Comments Or Suggestions:**

I have no further comments or suggestions besides what is mentioned above.

**Other Strengths And Weaknesses:**

**Strength**: I would like to emphasize that the paper presents interesting ideas and the theoretical results provided are nice to have.

**Weaknesses**: The paper could be challenging to follow for readers less familiar with sampling methods. For instance, providing a clear algorithm would help readers better understand and implement the proposed method. The paper also lacks explanations and discussions, and is poorly written at times.

**Questions For Authors:**

- Can you give explicit details about how you sample the conditional distributions? Do you use the discrete Langevin sampler?
- What happens in practice when you set $\alpha$ to be very close to $0$ and use a very large number of measurements?
- Is it useful for the two-stage sampler to start with a large $\eta$ and then decrease slowly decrease it?
- Once you decrease $\alpha$ significantly and then scale $m$ accordingly so that the denoiser performance is decent, the initial distribution of $(Y_1, \dotsc, Y_m)$ should be very close to a uniform. In this case, the 2-stage sampler would no longer be needed. Can the authors discuss this setting?

**Relation To Broader Scientific Literature:**

The paper builds on ideas present in the paper [1]. Still, there are significant differences in terms of scope (here discrete state space) and overall the methodology. The paper is also related to recent works on discrete diffusion which are cited in the paper. I believe that the contribution in this paper is novel wrt these prior works. However, besides emphasizing the difference in methodology with these works, the paper does not discuss the main practical interest of their method vs these prior works.

[1] Saremi, S., Park, J.W. and Bach, F., 2023. Chain of log-concave Markov chains. arXiv preprint arXiv:2305.19473.

**Theoretical Claims:**

The proofs are, as far as I am concerned, correct.

---

> ### Author Rebuttal · Authors · 2025-03-31
>
> Thank you for your review.
>
> Could the reviewer comment on "the authors proceed to develop a Gibbs sampler for the initial distribution"? The algorithm we have studied is not a Gibbs sampler.
>
> Regarding "In my opinion, the methodology lacks motivation", our motivation is that in many applications, a single noise level is enough, which in addition can be extended by adding a single hyperparameter (the number of measurements) to improve sample quality.
>
> Regarding "Essential References Not Discussed", we'd be happy to add the references mentioned by the reviewer.
>
> Regarding the comment that the paper "is poorly written at times", could the reviewer comment on where it is poorly written? We'd be happy to improve the presentation.
>
> Regarding the question on sampling from the conditional distributions, this is explained below Eq. 4. The conditional distributions also satisfy the assumptions (4), which essentially follow from the identities in Section 2. This is also quite intuitive, as sampling from conditional distributions should not be harder than sampling the first noisy measurement. Given this question by the reviewer, we will clarify this further in our revision.
>
> Regarding "Questions For Authors":
>
> - Yes for sampling from the measurement-conditioned distributions, we can use the discrete Langevin sampler.
>
> - Changing the step size is a good suggestion for future work. Here, we took a simple approach, choosing a fixed step size, which we also did not tune. The step size is simply set to $1/\alpha$.
>
> - This is an interesting regime. However, the tradeoff is that for small $\alpha$ sampling each measurement becomes easier but then one has to wait longer in measurement accumulation to have good denoised samples. For some problems even a single measurement may be enough, but then one shouldn't use a very small $\alpha$. We do not believe there is a general answer on which strategy is better as this is highly problem dependent.
>
> **GENERAL RESPONSE**:
>
> We would like to thank all the reviewers for their detailed feedback, highlighting the novelty and simplicity of our framework and providing pointers to the literature on discrete diffusion. We address the common concerns here.
>
> Three reviewers brought up references on discrete diffusion that we're happy to add to our literature review. We'd also be happy to extend our discussion of the diffusion approach vs. ours (non-SDE, single noise level) in the related work section.
>
> The closest reference to our paper is the concurrent paper by Le Tuyet et al., Discrete Markov Probabilistic Models (posted first on arXiv after the ICML deadline). We're happy to discuss this concurrent work as well. In short, their method is very different from ours, as it involves a "continuous-time Markov chain."
>
> Several reviewers commented on the need for pseudocode and schematics. Thank you for this constructive feedback. We will add pseudocode and a schematic in our revision.
>
> Regarding novelty, please note that there are no known results on the mixing time of discrete Langevin MCMC, and our convergence results also improve upon the known results [1]. In their paper, they assume the target distribution is quadratic (Eq. 5 in [1]), which we relax substantially with our assumption (4) in our paper.
>
> [1] Zhang et al. (2022) A Langevin-like Sampler for Discrete Distributions

---

> > ### Comment · Reviewer_REiK · 2025-04-03
> >
> > I would like to thank the authors for their response.
> >
> > I am not sure I understand what is wrong with the claim that the algorithm used to sample from the initial distribution is an (approximate) Gibbs sampler. The joint density $q(y, z)$ has $q$ as marginal wrt $y$ and the proposed algorithm proceeds by iteratively sampling each conditional, with one step (the conditional $y|z$) implemented approximately.
> >
> > > the methodology lacks motivation
> >
> > Overall, the steps of the algorithm are: first use a two-stage sampler for the initial distribution, then use discrete Langevin steps to   sample from each conditional distribution, then denoise. It feels comparable to the initial paper on Diffusion models [1] where the authors use Langevin dynamics to sample from the transitions because it was not known back then that we could come up with simpler and more principled updates that do not require significant parameter tuning, and obtain much better performance on top of that. The fact that the method in the present paper has to resort to a two-stage sampler for the initial distribution and discrete Langevin for the conditionals feels like a set back and I am still not convinced why one would use this method and not simply define a forward Markov chain of sign flips and then reverse the process. The initial distribution and the transitions would be much simpler to sample.
> >
> > I agree that one noise level may be enough in some applications and this would be appealing if the initial distribution is easy to sample. But this is not the case and so the solution is to either add many more measurements at the same noise level. You then end up with a sequence of transitions that you need to sample. It is thus not clear that your method is more compute efficient than simply reversing a Markov chain.
> >
> > Given these concerns, I maintain my score.
> >
> > [1] Song, Y. and Ermon, S., 2019. Generative modeling by estimating gradients of the data distribution.

---

> > > ### Author Response · Authors · 2025-04-09
> > >
> > > We appreciate your continued feedback.
> > >
> > > Indeed, we add a Gibbs sampler in the two-stage case, but sampling in {-1,1}^d is not done by coordinate-wise steps; all coordinates are updated simultaneously. We can make this clearer in the revision. Please also note that in the vanilla (single-stage) case, there is only one set of coordinates, and all are updated simultaneously using the score function. Our paper analyzes both single-stage and two-stage samplers.
> > >
> > > The setting in our work is very different from that in the paper the reviewer cited. Here, we are interested in sampling from a distribution on the Boolean hypercube, specifically in a setting where Gaussian noise is not allowed. We already have a discussion of the diffusion literature in the Gaussian case in the introduction, but we would be happy to extend that discussion, as we stated in our general response.
> > >
> > > The literature on discrete Langevin MCMC is very new, and as we have emphasized, there is simply no convergence analysis in the general case. Our paper addresses this with Propositions 3.1 and 3.3. In addition, we prove results regarding the mixing time of the discrete Langevin algorithms (Propositions 3.2 and 3.4). We believe that none of these results are related to the paper the reviewer cited.
> > >
> > > The paper most relevant to our work is the concurrent SDE-based work by Le Tuyet et al., which was cited by Reviewer Fbys. Compared to the concurrent work, we can make much bigger steps and require fewer score functions to learn (one with a single noise level, and only $m$ in the general case).
> > >
> > > Regarding models with a single noise level, we would also like to re-emphasize that there are cases where a single noise level is capable of achieving state-of-the-art results. We have already cited three such published papers in the introduction (by Pinheiro et al., Frey et al., and Kirchmeyer et al.). We'd be happy to extend that discussion and add more references.

---

### Official Review · Reviewer_XYSZ · 2025-03-12

**Overall Recommendation:** 4

**Summary:**

This paper investigates the problem of sampling from distributions over the binary hypercube using a smoothing-denoising framework. Instead of adding Gaussian noise, the authors introduce a novel approach based on Bernoulli noise. To enhance convergence in the denoising step, they leverage proximal sampling methods. The effectiveness of their method is demonstrated through experiments on synthetic data and binarized images.

**Claims And Evidence:**

The claims are generally well-supported by proofs and empirical experiments.

**Essential References Not Discussed:**

I did not find any essential related works that are missing.

**Experimental Designs Or Analyses:**

I am less familiar with the experimental aspects, so I only conducted a high-level review. Based on my assessment, the experimental design appears reasonable and generally supportive of the claims.

**Methods And Evaluation Criteria:**

The proposed methods make a significant contribution to sampling from Boolean distributions. While smoothing discrete distributions using Gaussian noise has been extensively studied, the use of Bernoulli noise appears to be a novel and promising approach. A key advantage of Bernoulli noise is its potential to improve mixing time. However, I am uncertain whether a similar convergence rate could be achieved with Gaussian noise when combined with a proximal sampler. Additionally, while I am less familiar with the experimental aspects, I would be interested in understanding the feasibility of applying this method to high-dimensional settings.

**Other Comments Or Suggestions:**

- Line 161 "is equal is" -> "is equal to"

**Other Strengths And Weaknesses:**

Strengths:

- this paper is well written;

- the use of Bernoulli noise instead of Gaussian noise is new for me;

**Questions For Authors:**

- In Section 4, can Gaussian noise with proximal sampling achieve a comparable convergence rate? If that is the case, the advantage of using Bernoulli noise might be limited.

- Is it possible to derive convergence results under other metrics, such as total variation (TV) distance?

- Could you provide a justification for the assumptions made in Section 3?

**Relation To Broader Scientific Literature:**

I don't know.

**Theoretical Claims:**

I checked the proofs and didn't find problems.

---

> ### Author Rebuttal · Authors · 2025-03-31
>
> Thank you for your review.
>
> Regarding "Other Comments Or Suggestions":
> - Thank you, we'll correct the typo.
>
> Regarding "Questions For Authors":
>
> - Thank you for the question. Intuitively, kinetic Langevin algorithms or Hamiltonian Monte Carlo also have an auxiliary variable ("velocity" in their case). However, making a deeper connection is a great research direction.
>
> - We believe more assumptions need to be made to extend our results to TV.
>
> - The assumptions made in Sec. 3 are akin to regularity assumptions (very common in the optimization literature) in the Euclidean case, which are adopted here for the binary case. In addition, they were required in our proofs in Sec. 3. We'd be happy to clarify this further in the paper.
>
> **GENERAL RESPONSE**:
>
> We would like to thank all the reviewers for their detailed feedback, highlighting the novelty and simplicity of our framework and providing pointers to the literature on discrete diffusion. We address the common concerns here.
>
> Three reviewers brought up references on discrete diffusion that we're happy to add to our literature review. We'd also be happy to extend our discussion of the diffusion approach vs. ours (non-SDE, single noise level) in the related work section.
>
> The closest reference to our paper is the concurrent paper by Le Tuyet et al., Discrete Markov Probabilistic Models (posted first on arXiv after the ICML deadline). We're happy to discuss this concurrent work as well. In short, their method is very different from ours, as it involves a "continuous-time Markov chain."
>
> Several reviewers commented on the need for pseudocode and schematics. Thank you for this constructive feedback. We will add pseudocode and a schematic in our revision.
>
> Regarding novelty, please note that there are no known results on the mixing time of discrete Langevin MCMC, and our convergence results also improve upon the known results [1]. In their paper, they assume the target distribution is quadratic (Eq. 5 in [1]), which we relax substantially with our assumption (4) in our paper.
>
> [1] Zhang et al. (2022) A Langevin-like Sampler for Discrete Distributions

---

> > ### Comment · Reviewer_XYSZ · 2025-04-05
> >
> > Thanks for your detailed response. Overall, I find this to be an interesting piece of work. I just wanted to follow up with a question regarding the two-stage sampler: what are the main challenges in applying a Gibbs-type sampler to the Gaussian smoothing case discussed in Section 4?

---

> > > ### Author Response · Authors · 2025-04-09
> > >
> > > We’re glad you found our work interesting.
> > >
> > > We are not entirely sure what the question is referring to. In the Gaussian case, the most natural "Gibbs-type" algorithm that extends the unadjusted Langevin algorithm is the kinetic Langevin MCMC, also known as underdamped Langevin MCMC. In that case, one augments the distribution by adding an auxiliary ("velocity") random variable and alternates between updating the velocity and then the coordinates (see Eq. 1 in [1]). This is conceptually similar to the two-stage sampler we use, but technically these algorithms are very distinct. In particular, there is a lot of freedom in discretizing the kinetic Langevin diffusion, which is a very active area of research. The introduction of [2] contains a good literature review on this topic.
> > >
> > > One could, in principle, consider a case where the auxiliary variables are discrete. This is an interesting research direction, if the reviewer has that setting in mind.
> > >
> > > [1] Cheng, X., Chatterji, N. S., Bartlett, P. L., & Jordan, M. I. (2018). Underdamped Langevin MCMC: A non-asymptotic analysis. In Conference on learning theory.
> > >
> > > [2] Mou, W., Ma, Y. A., Wainwright, M. J., Bartlett, P. L., & Jordan, M. I. (2021). High-order Langevin diffusion yields an accelerated MCMC algorithm. Journal of Machine Learning Research, 22(42), 1–41.

---

### Official Review · Reviewer_GyQ3 · 2025-03-14

**Overall Recommendation:** 3

**Summary:**

The authors propose a method to sample binary (vector) data by denoising through score functions. They first propose a noise model for binary data where the noise corresponds to a bit flipping with a given probability. Then, they construct the optimal denoiser based on Hamming loss and they show that it corresponds to a sign of a scaled score function (or a conditional expectation of clean data given the noise). Furthermore, they demonstrate better denoising performance provided that more noisy samples (of a fixed clean sample), are available. After that, they construct a sampling algorithm based on a two-stage discrete Langevin sampler (which employs Gibbs sampling), which allows them to produce more "noisy observations” of a fixed clean sample. The authors complete their study with experiments in the case of binarized MNIST.

**Claims And Evidence:**

* Claim: A model for noise for binary vectors based on flipping the bits with given probability. Evidence: mathematical derivations

* Claim: Optimal denoiser for this noise model given a Hamming loss which is given by a sign of a conditional expectation of clean data given a noise sample. Evidence: mathematical derivations and proofs

* Claim: Connection of a conditional expectation of clean data given a noise sample, to a score function of a noise model. Evidence: mathematical derivations.

* Claim: Approach to learn this score function via logistic regression. Evidence: Mathematical derivations.

* Claim: Approach to use multiple noisy samples for a one fixed clean sample which allows to better approximate the distribution of clean samples. Evidence: lemma

* Claim: Approach to use discrete Langevin algorithm to sample many noisy samples from a noise model. Evidence: derivations + proofs

* Claim: The proposed approach has better mixing times than Gaussian-based Langevin. Evidence: Derivations and proofs

* Claim: Empirical evidence that the proposed approach leads to a good approximation of a target distribution depending on the noise levels. Evidence: experiments.

* Claim: Empirical evidence that the approach works with Binarized MNIST. Evidence: Experiments.

I would like to challenge the claim that the approach of [1] is the first/only one to consider a generative model approach which does not rely on a SDE, as suggested by the end of the introduction "The main conceptual difference is that the multi-measurement sampling does not involve discretizing an SDE". The original formulation of DDPM, see [2,3] for instance, also does not include any SDE discretisation. Similarly, the fact that only one noise level is required is not a specificity of the given model. Such approaches are common in the image processing literature with the Plug-and-Play approaches [4].

Similarly, I think the claim that all approaches that build on diffusion models require a forward backward formulation (in the discrete setting as emphasized in Section 1.2) is a bit misleading. For example [5] is not based on a SDE perspective (I concede that they use multiple and different noise levels). In particular, there is no discretization approximation in this work.

[1] Saremi et al. (2024) Chain of log-concave Markov chains

[2] Sohl-Dickstein et al. (2015) Deep Unsupervised Learning using Nonequilibrium Thermodynamics

[3] Ho et al. (2020) Denoising Diffusion Probabilistic Models

[4] Romano et al. (2016) The Little Engine that Could: Regularization by Denoising (RED)

[5] Austin et al. (2020) Structured Denoising Diffusion Models in Discrete State-Spaces

**Essential References Not Discussed:**

I think a few recent references are missing.
Notably, there has been some great advances in the direction of discrete diffusion such as [1,2] and the references therein.
I think it would be worth emphasizing the connections between the current approach and these works.

[1] Shi et al. (2024) Simplified and Generalized Masked Diffusion for Discrete Data

[2] Zhao et al. (2024) Unified Discrete Diffusion for Categorical Data

**Ethical Review Concerns:**

No concerns

**Experimental Designs Or Analyses:**

Experiments are quite simple and straightforward, no concerns for validity.

**Methods And Evaluation Criteria:**

Most of the paper provides mathematical justifications together with error bounds for all the claims. Moreover, it provides empirical evidence highlighting all the theoretical claims made in the paper. The paper is mostly theoretical.

The experimental setup of the paper is however quite weak and does not emphasize the scalability of the method.

**Other Comments Or Suggestions:**

I suggest to the authors to add a discussion on how they see this work could scale to more complex discrete distributions (i.e., even just higher dimensional colored images?). What if the data is not binary, how would the authors approach it?
I appreciate that the authors added a discussion that this approach might be applicable to a broader set of exponential families distributions. It would be interesting to have a slightly more concrete discussion for a path forward.

**Other Strengths And Weaknesses:**

I think the paper is well written, all the claims are theoretically justified.
The method is a straightforward extension of [1] so the novelty is limited. However, I appreciate the current presentation.
The experiments are quite weak.

[1] Saremi et al. (2024) Chain of log-concave Markov chains

**Questions For Authors:**

Motivation: the motivation as to why consider only level of noise (even in the multisample case) is a bit unclear to me. One could say that we avoid the discretisation of the SDE but there is still some discrepancy between the target measure and the approximated one. Overall, I think that the authors should be able to answer the following question: why would a user pick this method over discrete diffusion or autoregressive models?

Scalability: How would it scale for non-binary discrete data and to a high dimensional colored images? The current experimental setup does not bring a lot of information regarding the efficiency of the method.

Baseline: no baseline is presented in this work. As it is presented as an alternative to discrete diffusion models it would be interesting to compare the performance in this case

**Relation To Broader Scientific Literature:**

This paper takes a very different approach to discrete modeling compared to Masked diffusion approach and discrete-from-continuous approaches.

**Theoretical Claims:**

I skimmed through the appendix and checked most of the proofs. Looks correct. Most of the theoretical claims have already been established in the literature. The main results of the paper in my opinion are Proposition 3.1 and Proposition 3.2.

---

> ### Author Rebuttal · Authors · 2025-03-31
>
> Thank you for your review.
>
> Please note in the original DDPM formulation, there is indeed an SDE behind the scenes. In other words, for understanding the theoretical properties of DDPM one has to return to the SDE formulation. The case of measurement accumulation is discrete by nature. Algorithmically, the methods are differentiated as in the DDPM case, one needs to come up with a forward and backward "noise schedule" and the case of measurement accumulation one has to only pick the number of measurements.
>
> Regarding the comment on denoising based on "one noise level", we're aware of the extensive research on denoising in the computer vision and signal processing literature, but turning that into a sampling and generative modeling scheme is a not as well studied. For example, the reference Romano et al. (2016) is not a generative model -- there is no connection between the score function and the denoiser in the paper.
>
> We're not sure what the reviewer means by "Most of the theoretical claims have already been established in the literature". The results in Sec. 2 are straightforward generalizations of the results for the Gaussian case. Conceptually, the emergence of a smooth score function that's defined beyond the Boolean hypercube is new. All the results in Sec. 3 are new -- in particular, there is no mixing time result for discrete Langevin MCMC in the literature. We'd be happy to clarify this further in our revisions.
>
> Regarding "few recent references are missing", we'd be happy to add the references mentioned by the reviewer in our revisions.
>
> We disagree that this paper is "straightforward extension of [1]". The machinery for discrete sampling and its theoretical properties have little in common with the Gaussian case. But conceptually, we also consider it as an extension of the the previous approach and in our view it's the strength of the method, since we arrive an algorithm with only two hyperparameters ($\alpha$ and $m$). Our
>
> [1] Saremi et al. (2024) Chain of log-concave Markov chains
>
> Regarding "Other Comments Or Suggestions":
>
> - We have an extension of the approach for non-binary data, but it was beyond of the scope of the paper due to space. One simple approach would be to use the 1-hot encoding for any discrete data that turns it into binary data.
>
> Regarding "Questions For Authors":
>
> - Regarding "why would a user pick this method over discrete diffusion", the choice really comes down to the fact for some applications, single noise models give rise to strong empriical results. Please see the references in the introduction. The models we have proposed in our paper are much simpler in terms of hyperparameters (and arguably in terms of formalism), compared to discrete diffusion. Please also note that we did not do any tuning for our experiments - the step size was simply set to 1/$\alpha$.
>
> - Our paper is mainly theoretical and the experiments were designed to understand the method. There are extensive comparisons between different regimes in our experiments. The experiments on MNIST are qualitative to show the fast mixing of our algorithm. For this, we encourage the reviewer to compare our results with the concurrent work by Le Tuyet et al., as mentioned by the reviewer Fbys - in our case only one or two steps are required to arrive at a digit. Also, we don't know how to adapt the diffusion approach to give single (fast-mixing) MCMC chains we demonstrated in our experiments.
>
> **GENERAL RESPONSE**:
>
> We would like to thank all the reviewers for their detailed feedback, highlighting the novelty and simplicity of our framework and providing pointers to the literature on discrete diffusion. We address the common concerns here.
>
> Three reviewers brought up references on discrete diffusion that we're happy to add to our literature review. We'd also be happy to extend our discussion of the diffusion approach vs. ours (non-SDE, single noise level) in the related work section.
>
> The closest reference to our paper is the concurrent paper by Le Tuyet et al., Discrete Markov Probabilistic Models (posted first on arXiv after the ICML deadline). We're happy to discuss this concurrent work as well. In short, their method is very different from ours, as it involves a "continuous-time Markov chain."
>
> Several reviewers commented on the need for pseudocode and schematics. Thank you for this constructive feedback. We will add pseudocode and a schematic in our revision.
>
> Regarding novelty, please note that there are no known results on the mixing time of discrete Langevin MCMC, and our convergence results also improve upon the known results [1]. In their paper, they assume the target distribution is quadratic (Eq. 5 in [1]), which we relax substantially with our assumption (4) in our paper.
>
> [1] Zhang et al. (2022) A Langevin-like Sampler for Discrete Distributions

---

### Official Review · Reviewer_Fbys · 2025-03-23

**Overall Recommendation:** 3

**Summary:**

The authors introduce a denoising sampling algorithm for distributions supported on the d-dimensional Boolean hypercube. In this discrete context, Gaussian noise does not exist and therefore one has to argue differently. The idea is to noise the target distribution by flipping each coordinate with a given probability, which depends upon a parameter $\alpha$. The smaller $\alpha$, the more noise is added to the target distribution. Next, leveraging a clean expression of an optimal denoiser through a conditional expectation, an approximate denoiser is learned from noisy samples using either logistic regression or a least-squares loss. Remarkably, the optimal denoiser admits a representation as the rescaled (continuous) gradient of the logarithm of the noisy distributions. This in parallel with well known results for Gaussian noising. A theoretical bound on the denoising performance is provided at Lemma 2.3. Section 2.4 explores adaptation of this framework to multiple measurements. That is, when the same sample is noised several time by the addition of i.i.d. noise. Section 3 considers the problem of sampling from the noisy distribution. First, a one-stage discrete Langevin sampler is proposed. Contractivity properties under assumptions on $\alpha$ of the sampler are proven in Proposition 3.1 and a control between the ergodic distribution of the sampler and the noisy distribution is the object of Proposition 3.2. Next, a two-stage Langevin sampler is considered and the same results as for the one-stage sampler are proven. The paper is concluded by a section that contains numerical experiments; first synthetic data are considered in the form of mixtures of two independent binary vectors. Then, the binary mist dataset is considered.

**Claims And Evidence:**

Yes

**Essential References Not Discussed:**

I am not aware of prior work that considers the same algorithm proposed in this work. I am aware of the concurrent work by Le Tuyet et al., which studies that same problem but from a different perspective. https://arxiv.org/abs/2502.07939 . There is a large literature on generative modeling based on Markov chains. I believe the paper "Discrete diffusion Schrodinger bridge matching for graph transformation" by Kim et al. published at ICLR2025 should enter the literature review.

**Experimental Designs Or Analyses:**

I checked the soundness of the experiments in section 5; they seem appropriate to me.

**Methods And Evaluation Criteria:**

Yes they are appropriate

**Other Comments Or Suggestions:**

* The authors should provide some schematic view or pseudo code for their algorithm(s)
* The authors shall provide an intuition about the proof of the result of section 3. Do they follow from known results?
* The authors should comment on why the parameter $\eta$ does not enter the bounds at Proposition 3.2 and 3.4 . It is not very clear to me

**Other Strengths And Weaknesses:**

Points of strength
* The paper addresses a fundamental problem, and I believe there is potential for several applications.
* The proposed algorithm is new, and offers a simpler framework than most previously generative algorithms for generating discrete sequences. The training objective is explicit, and offers an interesting parallelism with Gaussian denoising algorithms.


Weaknesses
* The proposed guarantees of convergence are weak. Lemma 2.3 is interesting as it makes no assumptions on the target $p$. But at the same time, the proposed bound is very rough. One would expect to see that mild regularity assumptions on p such as, say, finte entropy translate into better bounds. The authors should improve on this issue.

* There is essentially no precise discussions about the comparison with generative algorithms based on forward-backward Markov chains. This is true for both theory and experiments.

**Questions For Authors:**

1) The log-concavity estimate (8) tells that the modulus of log-concavity of $q$ is independent on the dimension $d$. Of course, this is false when noising in a Gaussian context.Can the authors comment on this? I suspect that this is because noising at level $\alpha$ in truth means that the computational cost of this operation is of order $\alpha d$. If so, it would be more correct to rewrite the results substituting $\alpha$ with $\alpha/d$.

2) I am confused from the statement right after Propostion 3.1. The authors seem to claim that the condition $\alpha\leq 1/(4\sqrt{d})$ implies the condition $4\alpha^2de^{2\alpha}\leq 1$. This claim seems false to me because of the term $e^{2\alpha}$ and if so it would have an important impact on the theoretical guarantees of the sampling part. It is likely that I am missing something obvious here, but I would still like the authors to clarify this point

**Relation To Broader Scientific Literature:**

The contributions of this article fit into the general literature about generative modeling. There has been already a significant number of papers that proposed score-based methods to sample from discrete data distributions. As the authors correctly report, most of these method use a forward-backward construction based on continuous-time diffusion processes, inspired by diffusion models. The approach undertaken here works in a single denoising step, and does not appeal to discretization of continuous-time processes. Citing this structural difference, the authors do not enter into any detailed comparison with prior work. I believe at least some comments on why doing all in a single step is worthwhile is needed in section 1.2

**Theoretical Claims:**

I checked the correctness of the proofs which are not in the appendix.

---

> ### Author Rebuttal · Authors · 2025-03-31
>
> Thank you for your review.
>
> Regarding "Essential References Not Discussed", the concurrent work by Le Tuyet et al. (the paper became public after the ICML submission deadline) is different from our paper, as it is based on a "continuous-time Markov chain," but both address the problem of sampling from a distribution on the Boolean hypercube. We'd be happy to cite the concurrent work as well as the paper by Kim et al. that the reviewer mentioned.
>
> Regarding "Weaknesses":
>  - Improving the bound in Lemma 2.3 is a great research direction, but we're not sure what the reviewer means by "finite entropy." Entropy is always finite in the discrete space.
>  - Regarding "precise discussions," please see our general response. In short, we'd be happy to extend our discussions of the diffusion literature.
>
> Regarding "Other Comments Or Suggestions":
>
> - Regarding "pseudo code," please see our general response.
>
> - The results in section 3 do not follow from known results.
>
> - The parameter $\eta$ does enter in Proposition 3.4 (see the condition in the proposition; $\eta$ cannot be too large, depending on $\beta_1$). The reviewer has a good point regarding $\eta$ not entering in the distance to the stationary distribution, which we'd be happy to emphasize too in our revision. But please note that, in both cases, based on Propositions 3.1 and 3.3, if $\eta$ is too small the mixing time will be too large, which is also quite intuitive.
>
> Regarding "Questions For Authors":
>
> - The main difference between the Bernoulli and the Gaussian case is that the the modulus of the random variables is constant on the Boolean hypercube. This may answer the reviewers question regarding "the modulus of log-concavity". We're not sure what the reviewer means by "computational cost" in this question. The motivation for swapping $\alpha$ with $\alpha/d$ is not clear either. The parameter $\alpha$ has the semantics of the noise parameter for each coordinate, and it's chosen due to its connection between $q_\alpha$ and the denoiser (the unnumbered equation in page 2).
>
> - Since $0 < 2\alpha \leq 0.5$, we have $e^{2\alpha} \leq 2$. It then follows that $4 \alpha^2 d e^{2\alpha} < 0.5$. We will clarify this in the paper.
>
> **GENERAL RESPONSE**:
>
> We would like to thank all the reviewers for their detailed feedback, highlighting the novelty and simplicity of our framework and providing pointers to the literature on discrete diffusion. We address the common concerns here.
>
> Three reviewers brought up references on discrete diffusion that we're happy to add to our literature review. We'd also be happy to extend our discussion of the diffusion approach vs. ours (non-SDE, single noise level) in the related work section.
>
> The closest reference to our paper is the concurrent paper by Le Tuyet et al., Discrete Markov Probabilistic Models (posted first on arXiv after the ICML deadline). We're happy to discuss this concurrent work as well. In short, their method is very different from ours, as it involves a "continuous-time Markov chain."
>
> Several reviewers commented on the need for pseudocode and schematics. Thank you for this constructive feedback. We will add pseudocode and a schematic in our revision.
>
> Regarding novelty, please note that there are no known results on the mixing time of discrete Langevin MCMC, and our convergence results also improve upon the known results [1]. In their paper, they assume the target distribution is quadratic (Eq. 5 in [1]), which we relax substantially with our assumption (4) in our paper.
>
> [1] Zhang et al. (2022) A Langevin-like Sampler for Discrete Distributions

---

### Decision · Program_Chairs · 2025-05-01

**Decision:**

Accept (poster)

**Comment:**

This paper proposes a generative model for binary data on the Boolean hypercube using a denoising approach. The data is corrupted via random bit flips, and a denoiser is trained to reverse this process. To improve performance, the authors introduce a multi-measurement strategy, where the denoiser receives multiple independent noisy versions of the same sample, enabling better reconstruction and which makes sampling easier. A Gibbs sampling procedure is developed to approximate the joint distribution of these noisy inputs. The approach is theoretically grounded through optimal denoising under Hamming loss and is validated with experiments on binarized MNIST.

Most reviewers weakly support the acceptance of the paper, and I align with the general consensus.

Most reviewers agree that the paper makes an interesting contribution to generative modeling for discrete data. However, several of them note that the contributions, theoretical or empirical, are modest, and most importantly, the paper lacks comparisons with existing baselines. In addition, some reviewers point out that the paper does not sufficiently position itself within the literature on discrete diffusion models. I agree with these concerns. In particular, I found that:
- The conditions imposed on $\beta_1,\beta_2$ are very restrictive.
- Even though the paper is mainly theoretical, the absence of comparisons with existing methods makes it difficult to assess the advantages and limitations of the proposed approach.
- While the paper includes many results, it does not present clear theoretical bounds on the final output of the proposed methods.

Even if the paper is accepted, I strongly encourage the authors to carefully consider these comments, as well as those from the reviewers, and address them in the final version of the manuscript.